# Mechanic: A Learning Rate Tuner

**Ashok Cutkosky**
Boston University
Boston, MA
ashok@cutkosky.com

**Aaron Defazio**
Meta, FAIR
New York, NY
adefazio@meta.com

**Harsh Mehta**
Google Research
Mountain View, CA
harshm@google.com

## Abstract

We introduce a technique for tuning the learning rate scale factor of any base optimization algorithm and schedule automatically, which we call MECHANIC. Our method provides a practical realization of recent theoretical reductions for accomplishing a similar goal in online convex optimization. We rigorously evaluate MECHANIC on a range of large scale deep learning tasks with varying batch sizes, schedules, and base optimization algorithms. These experiments demonstrate that depending on the problem, MECHANIC either comes very close to, matches or even improves upon manual tuning of learning rates.

## 1 Introduction

Modern deep learning is driven by first-order stochastic optimization algorithms. These are algorithms that are designed to solve the classical stochastic optimization problem:

$$\min F(\mathbf{x}) = \min \mathbb{E}_{\mathbf{z}}[f(\mathbf{x}, \mathbf{z})]$$

where $\mathbf{z}$ is a minibatch of examples, $\mathbf{x} \in \mathbb{R}^d$ is the model parameters, and $f$ is the loss incurred by using weights $\mathbf{x}$ on the minibatch $\mathbf{z}$. A first-order algorithm follows the protocol:

1. Output a $t$th iterate $\mathbf{x}_t$.
2. Sample a random minibatch $\mathbf{z}_t$.
3. Compute $\mathbf{g}_t = \nabla f(\mathbf{x}_t, \mathbf{z}_t)$ (the gradient is taken with respect to $\mathbf{x}_t$ only).
4. Possibly update some internal algorithm state based upon $\mathbf{g}_t$ in preparation for computing the next iterate $\mathbf{x}_{t+1}$.

The prototypical such optimization algorithm is stochastic gradient descent (SGD), which exemplifies the attractive features of this approach: it is computationally cheap (running in $O(d)$ time per update), and with proper tuning obtains minimax optimal convergence guarantees [1, 2]. Modern practice makes use of a wide range of variants of SGD, such SGD with momentum, AdaGrad [3], Adam [4], AdamW [5] or Lion [6]. Interest in such improvements to SGD is driven by the increasing computational demands of training large neural networks: better optimization means cheaper training, which translates to significant savings in terms of time, cost, and environmental impact.

Most modern algorithms for training neural networks are equipped with a scalar "scale factor" or "learning rate" hyperparameter $s \in \mathbb{R}$. Roughly speaking, these algorithms produce iterates of the form $\mathbf{x}_{t+1} = \mathbf{x}_t + s \cdot \mathbf{u}_t$ where $\mathbf{u}_t$ is some *update vector* produced as a function of the observed gradients $\mathbf{g}_1, \ldots, \mathbf{g}_t$ (we will use bold font for vectors in $\mathbb{R}^d$ like $\mathbf{u}$ and normal font for all other quantities like $s$). As an example, the classical SGD algorithm sets $\mathbf{u}_t = -\eta_t \mathbf{g}_t$ for some sequence of scalars $\{\eta_t\}$ typically called the *schedule*. The formula for the SGD update is:

$$\mathbf{x}_{t+1} = \mathbf{x}_1 - s \cdot \sum_{i=1}^{t} \eta_i \mathbf{g}_i. \tag{1}$$

The process of selecting the optimal $s$ is called "tuning", and is a key resource sink in machine learning. The typical approach is simply to try many possibilities to find the empirically optimal $s$, which requires multiple expensive training runs. This paper introduces a technique for choosing $s$ automatically on-the-fly in order to avoid this expense.

Our procedure, which we call MECHANIC, is a generic wrapper around any base optimization algorithm (BASE) that produces a new optimizer which does not require tuning of the scalar $s$. The base optimization algorithm is allowed to make any kind of update (for example, it may use any kind of schedule, preconditioner or weight decay). If $\mathbf{x}_t^{\text{BASE}} \in \mathbb{R}^d$ is the $t$th iterate of BASE, then the wrapper will produce a scalar $s_t \in \mathbb{R}$ and set the $t$th iterate of the wrapped algorithm to be $\mathbf{x}_t = \mathbf{x}_1^{\text{BASE}} + s_t(\mathbf{x}_t^{\text{BASE}} - \mathbf{x}_1^{\text{BASE}})$. As an example, suppose that BASE is the classical SGD algorithm with update equation (1). Then, given $s_t$, we would set $\mathbf{x}_t = \mathbf{x}_1^{\text{BASE}} - s_t \sum_{i=1}^{t-1} \eta_i \mathbf{g}_i$. Disregarding for now the fact that the gradients $\mathbf{g}_i$ actually depend on the iterates $\mathbf{x}_i$[1], we see that $\mathbf{x}_t$ is what the $t$th iterate of SGD *would have been* if the schedule were scaled by $s_t$.

Removing tuning of learning rate scalars is already a well-studied problem. One of the main attractions of early work in "adaptive" optimization such as AdaGrad and Adam [3, 7, 4] is that these algorithms require less tuning than ordinary SGD. Over the last decade, a number of works have aimed to tackle this problem from both an empirical and theoretical perspective [8, 9, 10, 11, 12, 13, 14, 15, 16, 17, 18, 19]. An intuitive approach might take the route of "hypergradient descent": that is, differentiating the update step of the optimization algorithm itself (e.g. [8, 9]). Strangely, it appears to be difficult to prove that such schemes behave well: theory-based approaches often adopt rather different strategies. Instead, we start from known theory and propose a few important modifications to produce a simple and effective practical implementation. We then rigorously evaluate our algorithm on a variety of datasets. We emphasize that our primary contribution is not new theoretical development, but instead the translation between theory and practice, which involves fusing several known analytical techniques as well as subtle departures from theory.

Previous works that investigate deep learning performance of "learning-rate free" optimization inspired by theory (e.g. [20, 21, 16, 15]) have already demonstrated impressive results. However, these works typically build "hand-crafted" algorithms that often blend theoretical analysis with specific empirically successful algorithms such as Adam. In contrast, our wrapper works well with any base algorithm and so can seamlessly integrate new empirical advances in optimization: one does not need intimate familiarity with the analysis of our approach to apply it to a new algorithm.

## 2 Background: Online Convex Optimization

We develop our formalism via *online convex optimization* (OCO) [22, 23, 24]. OCO is a popular framework for design and analysis of stochastic optimization algorithms. In brief, for each of $T$ rounds (corresponding to $T$ iterations of optimization), the OCO algorithm must first output a $t$th iterate $\mathbf{x}_t$, after which the algorithm is presented with a $t$th loss function $\ell_t$. Typically, one envisions the case $\ell_t(\mathbf{x}) = \ell(\mathbf{x}, \mathbf{z}_t)$ for some fixed loss $\ell$ and new data point $\mathbf{z}_t$. The goal of an algorithm ALG is to minimize the *regret* $R^{\text{ALG}}(\mathring{\mathbf{x}})$:

$$R^{\text{ALG}}(\mathring{\mathbf{x}}) \triangleq \sum_{t=1}^{T} \ell_t(\mathbf{x}_t) - \ell_t(\mathring{\mathbf{x}}).$$

Many references focus primarily on the case $\mathring{\mathbf{x}} = \operatorname{argmin} \sum_{t=1}^{T} \ell_t(\mathbf{x})$ in order to consider the single scalar value $\sup_{\mathring{\mathbf{x}}} R_T(\mathring{\mathbf{x}})$ [25, 26], but we will employ the formulation of regret as a function above instead as it is strictly more general. When $\ell_t$ is convex, then with $\mathbf{g}_t \triangleq \nabla \ell_t(\mathbf{x}_t)$ (or, more generally when $\mathbf{g}_t$ is a subgradient of $\ell_t$ at $\mathbf{x}_t$), we have:

$$R^{\text{ALG}}(\mathring{\mathbf{x}}) \leq \sum_{t=1}^{T} \langle \mathbf{g}_t, \mathbf{x}_t - \mathring{\mathbf{x}} \rangle \triangleq R_{\text{linear}}^{\text{ALG}}(\mathring{\mathbf{x}}).$$

As a result, the vast majority of OCO algorithms provide analysis that bounds only the linearized regret $R_{\text{linear}}^{\text{ALG}}(\mathring{\mathbf{x}})$. Such algorithms do not need to observe the entire function $\ell_t$: instead, they only make use of the gradients $\mathbf{g}_t$. That is, the $t$th output of ALG (i.e. $\mathbf{x}_t$) is purely a function of the previous sequence of gradients $\mathbf{g}_1, \ldots, \mathbf{g}_{t-1}$ so that ALG is a first-order algorithm.

---

[1]This seems like a significant issue to disregard, but we will provide mathematical justification shortly.

## 2.1 Learning the Scale in OCO

Just like stochastic optimization algorithms, most OCO algorithms also require a scale factor $s$. In fact, many stochastic optimization algorithms (such as SGD and AdaGrad) are *also* OCO algorithms. Setting $\eta_t = \eta$ for all $t$, SGD ensures the regret bound:

$$R^{\text{SGD}}(\mathring{\mathbf{x}}) \le R^{\text{SGD}}_{\text{linear}}(\mathring{\mathbf{x}}) \le O\left(\frac{\|\mathring{\mathbf{x}} - \mathbf{x}_1\|^2}{s\eta} + s\eta \sum_{t=1}^{T} \|\mathbf{g}_t\|^2\right). \tag{2}$$

From this equation, one can deduce in hindsight that for any given $\mathring{\mathbf{x}}$, the optimal value for $s$ is $\frac{\|\mathring{\mathbf{x}} - \mathbf{x}_1\|}{\eta\sqrt{\sum_{t=1}^{T}\|\mathbf{g}_t\|^2}}$, which would provide the bound:

$$R^{\text{SGD WITH TUNED } s}_{\text{linear}}(\mathring{\mathbf{x}}) \le O\left(\|\mathring{\mathbf{x}} - \mathbf{x}_1\| \sqrt{\sum_{t=1}^{T} \|\mathbf{g}_t\|^2}\right). \tag{3}$$

This result is minimax optimal [27], but requires knowledge of the unknown optimal $s$. Very recently, [14, 16, 15] have produced algorithms that estimate the value of $\|\mathbf{x}_1^{\text{BASE}} - \mathring{\mathbf{x}}\|$ on-the-fly and use this estimate to quickly identify the optimal scaling value $s$. These algorithms achieve impressive practical performance, but they require an understanding of the closed-form solution for the optimal $s$ value above. Our goal is to learn the correct scaling regardless of the base algorithm.

To this end, we will leverage a scheme recently developed by [28] that allows one to automatically tune the scale of a base OCO algorithm using another "meta" OCO algorithm. We reproduce their result below (with notation altered to suit our application) along with the short proof:

**Theorem 1** ([28]). *Suppose* BASE *and* TUNER *are both OCO algorithms. Let* $\{\mathbf{x}_t^{\text{BASE}}\} \subset \mathbb{R}^d$ *indicate the iterates of* BASE *in response to an arbitrary sequence of gradients* $\{\mathbf{g}_t\}$*, and let* $\{s_t\} \subset \mathbb{R}$ *indicate the iterates of* TUNER *in response to the sequence of scalars* $\{h_t = \langle \mathbf{g}_t, \mathbf{x}_t^{\text{BASE}} - \mathbf{x}_1 \rangle\}$*. Define a new online algorithm* MECHANIC *via:*

$$\mathbf{x}_t^{\text{MECHANIC}} = \mathbf{x}_1^{\text{BASE}} + s_t \cdot (\mathbf{x}_t^{\text{BASE}} - \mathbf{x}_1^{\text{BASE}}).$$

*Then* $\mathbf{x}_t^{\text{MECHANIC}}$ *ensures regret:*

$$R^{\text{MECHANIC}}_{\text{linear}}(\mathring{\mathbf{x}}) \le \inf_{\mathring{s}} R^{\text{TUNER}}_{\text{linear}}(\mathring{s}) + \mathring{s} R^{\text{BASE}}_{\text{linear}}((\mathring{\mathbf{x}} - \mathbf{x}_1^{\text{BASE}})/\mathring{s}).$$

*Proof.* By definition, for any $\mathring{s}$, we have:

$$\begin{aligned}
R^{\text{MECHANIC}}_{\text{linear}}(\mathring{\mathbf{x}}) &= \sum_{t=1}^{T} \langle \mathbf{g}_t, \mathbf{x}_1^{\text{BASE}} + s_t \cdot (\mathbf{x}_t^{\text{BASE}} - \mathbf{x}_1^{\text{BASE}}) - \mathring{\mathbf{x}} \rangle \\
&= \sum_{t=1}^{T} \langle \mathbf{g}_t, \mathbf{x}_t^{\text{BASE}} - \mathbf{x}_1^{\text{BASE}} \rangle (s_t - \mathring{s}) + \mathring{s} \sum_{t=1}^{T} \langle \mathbf{g}_t, \mathbf{x}_t^{\text{BASE}} - \mathbf{x}_1^{\text{BASE}} - (\mathring{\mathbf{x}} - \mathbf{x}_1^{\text{BASE}})/\mathring{s} \rangle \\
&= R^{\text{TUNER}}_{\text{linear}}(\mathring{s}) + \mathring{s} R^{\text{BASE}}_{\text{linear}}(\mathbf{x}_1^{\text{BASE}} + (\mathring{\mathbf{x}} - \mathbf{x}_1^{\text{BASE}})/\mathring{s}).
\end{aligned}$$

$\square$

With this result, the job of finding the optimal $s$ can usually be completely relegated to TUNER. Although the value $\mathring{s}$ appears in both terms of the sum $R^{\text{TUNER}}_{\text{linear}}(\mathring{s}) + \mathring{s} R^{\text{BASE}}_{\text{linear}}(\mathbf{x}_1^{\text{BASE}} + (\mathring{\mathbf{x}} - \mathbf{x}_1^{\text{BASE}})/\mathring{s})$, it turns out that for essentially all plausible BASE algorithms, there is a particular value $\mathring{s}$ that causes $\mathring{s} R^{\text{BASE}}_{\text{linear}}(\mathbf{x}_1^{\text{BASE}} + (\mathring{\mathbf{x}} - \mathbf{x}_1^{\text{BASE}})/\mathring{s})$ to obtain the optimal regret bound (3). Thus, by setting $\mathring{s}$ to be this value, which is unknown *a priori*, we need only ensure that $R^{\text{TUNER}}_{\text{linear}}(\mathring{s})$ is small enough to not significantly affect the overall regret bound. Note that this setting of $\mathring{s}$ is done entirely in the analysis. For example, if BASE is actually SGD with a learning rate $\eta$ and $s = 1$ as in (2), we have

$$R^{\text{MECHANIC}}(\mathring{\mathbf{x}}) \le R^{\text{MECHANIC}}_{\text{linear}}(\mathring{\mathbf{x}}) \le \inf_{\mathring{s}} R^{\text{TUNER}}_{\text{linear}}(\mathring{s}) + O\left(\frac{\|\mathring{\mathbf{x}} - \mathbf{x}_1\|^2}{\mathring{s}\eta} + \mathring{s}\eta \sum_{t=1}^{T} \|\mathbf{g}_t\|^2\right),$$

setting $\mathring{s} = \frac{\|\mathring{\mathbf{x}} - \mathbf{x}_1\|}{\eta\sqrt{\sum_{t=1}^{T}\|\mathbf{g}_t\|^2}}$:

$$\leq R_{\text{linear}}^{\text{TUNER}}\left(\frac{\|\mathring{\mathbf{x}} - \mathbf{x}_1\|}{\eta\sqrt{\sum_{t=1}^{T}\|\mathbf{g}_t\|^2}}\right) + O\left(\|\mathring{\mathbf{x}} - \mathbf{x}_1\|\sqrt{\sum_{t=1}^{T}\|\mathbf{g}_t\|^2}\right).$$

Thus, if TUNER obtains low regret, then we will obtain the same regret bound as if we had optimally tuned the scaling factor for SGD. Intuitively, the gradient $h_t$ provided to TUNER approximates the gradient over the entire course of the base optimizer rather than just at the most recent iterate. That is, for SGD, $h_t \approx \frac{df(\mathbf{x}_t, \mathbf{z}_t)}{ds}$ where $\mathbf{x}_t = \mathbf{x}_1 - s\sum_{k=1}^{t-1}\eta_k\mathbf{g}_k$.

## 2.2 Parameter-Free Online Optimization

The problem with the above result is that we seem to have simply pushed the problem off to TUNER: what if TUNER itself requires us to set a scale factor? Solving this problem has been the focus of a substantial effort in the online optimization community [29, 30, 10, 11, 28, 12]. The most advanced such algorithms are able to ensure for all $\mathring{s}$ simultaneously:

$$R_{\text{linear}}(\mathring{s}) = \sum_{t=1}^{T} h_t(s_t - \mathring{s}) \leq \tilde{O}\left(|\mathring{s}|\sqrt{\sum_{t=1}^{T}h_t^2}\right). \tag{4}$$

Thus, if we set $h_t = \langle\mathbf{g}_t, \mathbf{x}_t^{\text{BASE}} - \mathbf{x}_1^{\text{BASE}}\rangle$, we obtain:

$$R_{\text{linear}}^{\text{TUNER}}(\mathring{s}) \leq \tilde{O}\left(|\mathring{s}|\sqrt{\sum_{t=1}^{T}\langle\mathbf{g}_t, \mathbf{x}_t^{\text{BASE}} - \mathbf{x}_1^{\text{BASE}}\rangle^2}\right).$$

In a theoretical development of this technique, it is necessary to prevent the terms $\langle\mathbf{g}_t, \mathbf{x}_t^{\text{BASE}} - \mathbf{x}_1^{\text{BASE}}\rangle^2$ from becoming too large (as otherwise $R_{\text{linear}}^{\text{TUNER}}$ is too large). Typically, this is accomplished by constraining the base algorithm to satisfy $\|\mathbf{x}_t^{\text{BASE}} - \mathbf{x}_1^{\text{BASE}}\| \leq \rho$ for some user-specified arbitrary $\rho$. Enforcing such a constraint means that the regret bound (2) would only apply to $\|\mathring{\mathbf{x}}\| \leq \rho$, but ensures that $\langle\mathbf{g}_t, \mathbf{x}_t^{\text{BASE}} - \mathbf{x}_1^{\text{BASE}}\rangle^2 \leq \rho^2\|\mathbf{g}_t\|^2$. Thus, by setting $\mathring{s} = \|\mathring{\mathbf{x}} - \mathbf{x}_1^{\text{BASE}}\|/\rho$, the combined algorithm obtains the optimal regret bound of $O(\|\mathring{\mathbf{x}} - \mathbf{x}_1^{\text{BASE}}\|\sqrt{\sum_{t=1}^{T}\|\mathbf{g}_t\|^2})$ (amazingly, the value of $\rho$ is irrelevant!). In practice however, we do not attempt to explicitly enforce any such constraints and simply rely on the intuition that any non-diverging algorithm is unlikely to produce excessively large iterates.

At no point in this process do we need access to the internal state of the base algorithm BASE. This means that improvements to BASE will automatically be reflected in improvements to the overall algorithm. In this paper, we investigate the performance of MECHANIC on deep learning tasks. We consider a variety of settings for the base algorithm BASE (i.e. AdamW, Lion, SGD, with various batch sizes and learning rate schedules of various shapes), and employ a parameter-free algorithm as the TUNER to automatically find the best scale factor for the base algorithm.

## 3  The MECHANIC algorithm

Our MECHANIC algorithm is specified in Algorithm 1. The algorithm is built by applying Theorem 1 to a parameter-free TUNER algorithm presented in Algorithm 2, which is described along with theoretical analysis in Appendix D. However, when building MECHANIC, we modify the "pure" theoretically tractable Algorithm 2 to simplify the implementation while still capturing the essential intuition and maintaining the same performance. In the remainder of this section we will provide some intuition behind the TUNER update as used in MECHANIC as well as describing some potentially unfamiliar subtleties relating to our use of exponentially weighted moving averages.

MECHANIC takes as input a base algorithm that generates *update vectors* $\mathbf{u}_t$ as described in the previous sections. We then set $\mathbf{\Delta}_{t+1} = \sum_{k=1}^{t}\mathbf{u}_k = \mathbf{x}_{t+1}^{\text{BASE}} - \mathbf{x}_1^{\text{BASE}}$. The majority of the algorithm contains our TUNER method, which is a variant of the analytically tractable Algorithm 2, with a

**Algorithm 1** MECHANIC

1: **Input:** Base algorithm BASE, $\beta \in [0,1]^n$ (default $n = 6$, $\beta = (0.9, 0.99, 0.999, 0.9999, 0.99999, 0.999999)$, $\lambda \in \mathbb{R}$ (default $\lambda = 0.01$). $s_{init} \in \mathbb{R}$: first non-zero $s$ value (default $s_{init} = 10^{-8}$). $\epsilon = 10^{-8}$: small value for numerical stability.

2: $v_0 \leftarrow 0 \in \mathbb{R}^n, r_0 \leftarrow 0 \in \mathbb{R}^n, m_0 \leftarrow 0 \in \mathbb{R}^n, \mathbf{x}_{ref} \leftarrow \mathbf{x}_1^{\text{BASE}}$.

3: $\boldsymbol{\Delta}_1 \leftarrow 0 \in \mathbb{R}^d$

4: $s_1 \leftarrow 0 \in \mathbb{R}^n$. // We will use $s_{t,i}$ to indicate the $i$th coordinate of $s_t$.

5: **for** $t = 1 \ldots T$ **do**

6:     $\mathbf{g}_t \leftarrow \nabla f(\mathbf{x}_t, \mathbf{z}_t)$. // $\mathbf{x}_t$ is the $t$th set of model parameters and $\mathbf{z}_t$ is the $t$th minibatch.

7:     Send $\mathbf{g}_t$ to BASE, receive update $\mathbf{u}_k$. // On its own, BASE would update $\mathbf{x}_{t+1} \leftarrow \mathbf{x}_t + \mathbf{u}_k$.

8:     [Optional] Set $\boldsymbol{\Delta}_t = \frac{\mathbf{x}_t - \mathbf{x}_{ref}}{(\sum_{i=1}^n s_{t,n}) + \epsilon}$ to save memory instead of storing $\boldsymbol{\Delta}_t$ from last round.

9:     $\boldsymbol{\Delta}_{t+1} \leftarrow \boldsymbol{\Delta}_t + \mathbf{u}_t$.

10:     $h_t \leftarrow \left\langle \boldsymbol{\Delta}_t, \mathbf{g}_t + \frac{\lambda(\sum_{i=1}^n s_{t,n}) \|\mathbf{g}_t\| \mathbf{x}_t}{\|\mathbf{x}_t\|} \right\rangle$ // Note use of $\boldsymbol{\Delta}_t$ rather than $\boldsymbol{\Delta}_{t+1}$.

11:     $m_t \leftarrow \max(\beta m_{t-1}, h_t)$ (multiplications by $\beta$ and maximum are taken coordinate-wise)

12:     $v_t \leftarrow \beta^2 v_{t-1} + h_t^2$

13:     $r_t \leftarrow \beta r_{t-1} - s_{t-1} h_t$

14:     $r_t \leftarrow \max(0, r_t)$ // This step is used instead of more complicated procedures in Algorithm 2

15:     $W_t \leftarrow \frac{s_{init} \cdot m_t}{n} + r_t$

16:     $s_{t+1} \leftarrow \frac{W_t}{\sqrt{v_t} + \epsilon}$

17:     $\mathbf{x}_{t+1} \leftarrow \mathbf{x}_1^{\text{BASE}} + (\sum_{i=1}^n s_{t+1,i}) \cdot \boldsymbol{\Delta}_{t+1}$

18: **end for**

few modifications. Note that the indexing on $\boldsymbol{\Delta}$ is very important and may be counterintuitive: the definition of $h_t$ does *not* include $\boldsymbol{\Delta}_{t+1}$, but rather $\boldsymbol{\Delta}_t$. $h_t$ is the "gradient" that is supplied to TUNER, as described by Theorem 1.

To gain some intuition behind the update, let us consider the case that $n = 1$ and $\beta = 1.0$ (that is, without employing any exponentially-weighted moving averages). We keep track of the quantity $W_t = s_{init} \cdot m_t - \sum_{k=1}^t h_k s_k$, which is usually called the "wealth" of the algorithm (the quantity $r_t = -\sum_{k=1}^t h_k s_k$ is sometimes called the "reward"). $s_{init}$ specifies the starting value for $s_t$ and should be an under-estimate of the true optimal scaling. We then set $s_{t+1} = \frac{W_t}{\sqrt{v_t}}$ (neglecting the $\epsilon$ included for numerical stability). To understand this update strategy, we can re-write the update as:

$$s_{t+1} = s_t \cdot \frac{\sqrt{v_{t-1}}}{\sqrt{v_t}} - \frac{s_t h_t}{\sqrt{v_t}} \approx \left(1 - \frac{h_t^2}{2v_t}\right) s_t - \frac{s_t h_t}{\sqrt{v_t}}.$$

Thus, the update looks like a combination of an AdaGrad-esque gradient descent step with learning rate scaled by $s_t$ and a kind of "adaptive decay" (multiplication by $1 - \frac{h_t^2}{2v_t}$). The adaptive decay is very important for stabilizing the algorithm: without it the values for $s_t$ are prone to unchecked exponential growth due to scaling by $s_t$ in $\frac{s_t h_t}{\sqrt{v_t}}$. Intuitively, this decay is the minimum amount required to prevent instabilities.

In Appendix D, we provide a formal Theorem bounding the regret of a variant of the procedure described above. Roughly speaking, for $\beta = 1$ this result suggests:

$$\sum_{t=1}^T h_t(s_t - \mathring{s}) \leq O\left((\mathring{s} + \max_t s_t) \cdot m_T + \mathring{s} \cdot \log(T\mathring{s}/s_{init}) \sqrt{\sum_{t=1}^T h_t^2}\right). \tag{5}$$

In fact, the dependence of $O(\log(T))$ in equation (5) can be improved to $O(\sqrt{\log(T)})$ via more complicated algorithms (e.g. [28, 12, 31, 32]). However, we favor the simpler update and pleasing resemblance to familiar algorithms like AdaGrad via the Taylor expansion analysis above. Of note, the dependence on $s_{init}$ is very mild: this suggests that we should be able to set $s_{init}$ to a very small

value without damaging performance. In practice, we choose $s_{init} = 10^{-8}$, which we expect to dramatically underestimate the optimal value in all scenarios.

We hypothesize that the simplified TUNER we use in MECHANIC in fact possesses a rigorous theoretical analysis (although perhaps only with respect to simpler non-fully-worst-case adversaries), but demonstrating such a bound appears to involve difficult technical hurdles. In particular, our implementation is designed to be "scale-free": rescaling the values of $\mathbf{g}_t$ by any constant scalar will have no effect on $s_t$. This property was first achieved only recently in theoretical analysis of parameter-free algorithms [12], and as-yet requires significantly more involved algorithms [12, 33].

## 3.1 The use of $\beta$

We include $\beta$ to introduce some recency bias in the statistics recorded by MECHANIC, a common feature of practical optimizers. Mathematically, we accomplish this by up-weighting the $t$th feedback to TUNER by $\beta^{-t}$: $h_t \rightarrow h_t\beta^{-t}$. Thus, for example, we have $\mathbf{v}_t = \sum_{k=1}^t h_k^2 \beta^{-2kt}$ and $r_t = -\sum_{k=1}^t h_k s_{k-1}\beta_s^{-k}$. Using these weights directly results in numerical stability issues as the weights become exponentially large. Instead, since we only need to maintain the correct ratio $\frac{W_t}{\sqrt{v_t}}$, we can cancel a factor of $\beta_s^{-t}$ from both sides, giving the update equations in Algorithm 2.

We found that tuning the value of $\beta$ can significantly improve performance on different tasks. Thus, we incorporated multiple $\beta$ values simultaneously in a way that obviates the need for such tuning.

Our approach is inspired by work on "combining" parameter free algorithms [34]. The idea is simple: parameter-free algorithms typically ensure $R_{\text{linear}}(0) \leq \epsilon$ for some *constant* $\epsilon$ set by the user. So, if $s_{t,1}, \ldots, s_{t,n}$ are the outputs of $n$ parameter-free algorithms with regret bounds $R_{\text{linear}}^1(\mathring{s}), \ldots, R_{\text{linear}}^n(\mathring{s})$, we have for any $j$:

$$\sum_{t=1}^T h_t \left(\sum_{i=1}^n s_{t,i} - \mathring{s}\right) = \sum_{t=1}^T h_t(s_{t,j} - \mathring{s}) + \sum_{i\neq j}\sum_{t=1}^T h_t(s_{t,i} - 0)$$

$$= R_{\text{linear}}^j(\mathring{s}) + \sum_{i\neq j} R_{\text{linear}}^i(0) \leq R_{\text{linear}}^j(\mathring{s}) + (n-1)\epsilon.$$

So, with small constant additive overhead in the regret, the sum of all the outputs $s_{t,1} + \cdots + s_{t,n}$ achieves the same regret as the *best* of all the outputs. Motivated by this observation, we instantiate $n = 6$ copies of TUNER with different $\beta$ values and add their iterates to produce a final scaling.

## 3.2 Weight decay

Finally, we found that an addition of a peculiar weight-decay-esque term helped significantly on certain tasks, including vision tasks with smaller datasets, multi-objective NLP tasks and especially with reducing the variance in final results for all tasks. Specifically, rather than providing $h_t = \langle\mathbf{g}_t, \mathbf{\Delta}_t\rangle$ as the input to the TUNER algorithm, we instead provide $h_t = \langle\mathbf{g}_t + \frac{\lambda\|\mathbf{g}_t\|\left(\sum_{i=1}^n s_{t,i}\right)\mathbf{x}_t}{\|\mathbf{x}_t\|}, \mathbf{\Delta}_t\rangle$. We conjecture that this term is helpful in the common case that the base algorithm itself is incorporating regularization or weight-decay.

This extra term is the derivative of the regularizer $\mathbf{x} \mapsto \lambda\|\mathbf{g}_t\|\left(\sum_{i=1}^n s_{t,i}\right)\|\mathbf{x}\|$. From a standard theoretical perspective, this regularization may seem overly large. However, it may not have as big an impact as one might imagine because the base algorithm does not see this regularization. Instead, the base algorithm may (or may not) perform weight decay using another method that MECHANIC has no insight into. That said, we do not propose an analytical explanation for this modification. We simply observed that in practice it performed well with a fixed $\lambda = 0.01$.

## 3.3 Runtime and Memory Cost

MECHANIC incurs little additional cost over that of BASE. In Algorithm 1, we denote $d$-dimensional vectors with bold font, and $n$-dimensional vectors and scalars with normal font (note that typically $n = 6$). We use 1 additional $O(d)$ memory slot, and four $O(d)$-time steps in lines 8, 9, 10 and 17. All other steps are $O(1)$ or $O(n)$ time and so have negligible cost.

| Model | Size | Pre Opt | MLM | Optimizer | MNLI-m/mm | QNLI | SST-2 | QQP |
|---|---|---|---|---|---|---|---|---|
| BERT-B | 110M | AdamW | 71.5 | AdamW | 84.3/84.8 | 91.0 | 92.4 | 90.1 |
| | | | | $\mathcal{M}$-AdamW | 83.7/83.5 | 90.6 | 91.9 | 90.5 |
| | | $\mathcal{M}$-AdamW | **71.7** | AdamW | **84.7/85.1** | 91.2 | **93.3** | 90.7 |
| | | | | $\mathcal{M}$-AdamW | 84.5/84.4 | **91.3** | 92.5 | **91.1** |
| BERT-B | 110M | Lion | 71.8 | Lion | 83.4/83.5 | 86.8 | 89.7 | 89.4 |
| | | | | $\mathcal{M}$-Lion | 83.1/83.8 | **89.9** | 91.0 | 90.2 |
| | | $\mathcal{M}$-Lion | **72.0** | Lion | **84.5**/84.2 | 89.0 | **91.2** | **90.8** |
| | | | | $\mathcal{M}$-Lion | 84.2/**84.2** | 88.6 | 91.1 | 90.2 |
| BERT-L | 340M | AdamW | **75.4** | AdamW | 86.2/86.4 | 92.2 | 93.9 | 91.3 |
| | | | | $\mathcal{M}$-AdamW | 86.1/86.4 | 92.5 | 93.7 | 91.5 |
| | | $\mathcal{M}$-AdamW | 75.3 | AdamW | **86.3/86.5** | **92.7** | **94.4** | 91.4 |
| | | | | $\mathcal{M}$-AdamW | 86.1/86.3 | 91.7 | 93.5 | **91.5** |
| BERT-L | 340M | Lion | **75.7** | Lion | 86.7/86.6 | 90.7 | 92.9 | 91.1 |
| | | | | $\mathcal{M}$-Lion | 86.0/86.2 | 90.3 | **93.4** | 91.2 |
| | | $\mathcal{M}$-Lion | 75.5 | Lion | **87.4/87.4** | **92.9** | 93.3 | **91.7** |
| | | | | $\mathcal{M}$-Lion | 87.2/87.1 | 91.5 | 92.3 | 91.6 |

Table 1: Comparing MECHANIC on BERT. 5 largest datasets from GLUE. Results reported are peak validation scores averaged over 3 runs, both for the baseline and MECHANIC tuned models.

## 4 Experiments

In this section we describe our experiments using MECHANIC to tune various base optimizers on different tasks. Note that almost all base optimizer implementations require a user-specified scale factor which is is not directly visible to MECHANIC. We set this value to 1.0 before applying MECHANIC. Since MECHANIC multiplies the base update by $s_t$, setting the base scale factor to 1.0 allows us to interpret $s_t$ as the "correct" value for the base scale.

### 4.1 Masked Language Modeling

We perform BERT pre-training on the Wikibooks dataset following the procedure from [35] with a few minor changes, most notably, we omit the Next Sentence Prediction (NSP) loss following [36]. Masked language modeling (MLM) requires reconstructing randomly masked tokens given an input sequence of tokens. As shown in Table 1, using MECHANIC leads to a noticeable improvement in MLM accuracy.

**Varying batch size and model size:** Past works observe that the scale factor $s$ should decrease as either batch size is decreased or model size is increased [37, 38]. To inspect the scale factor that MECHANIC learns, we vary the batch size and model size while pre-training BERT using ME-CHANIC. As shown in Figure 1, in both cases, MECHANIC learns to decrease the scale factor $s$ when decreasing the batch size and when increasing the model size.

**Addition Ablations:** Ablation studies on the effects of $n$, $\lambda$, $s_{init}$ can be found in Appendix B.

**Finetuning pre-trained models:** In addition to pre-training, we evaluate our models on the 5 largest datasets from the GLUE suite [39]. One possible failure mode of MECHANIC tuned pre-trained models could have been that, even though they lead to high accuracy at pre-training time, transfer learning may fail at finetuning time.

To ensure that standard transfer learning pipelines still work with MECHANIC pre-trained models, we finetune them without a learning rate tuner using the AdamW optimizer and find that MECHANIC pre-trained models lead to higher accuracy at pre-training time, and they also outperform in finetuning more often than not. We finetune BERT-B (110M) and BERT-L (340M) models for at most 10 epochs on each of the GLUE datasets and report results on the GLUE dev set in Table 1.

**Using MECHANIC for finetuning:** We also investigated using MECHANIC for finetuning. Typically, to not erase the progress already made, a much lower base learning rate is employed at finetuning

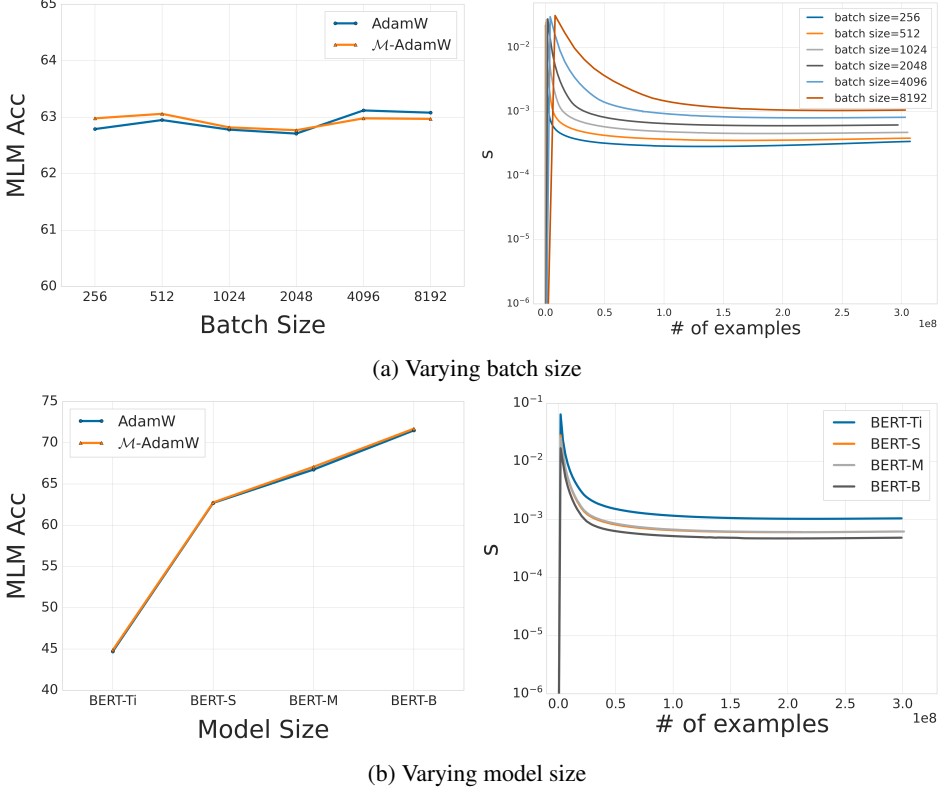

(a) Varying batch size

(b) Varying model size

Figure 1: Scaling values $s$ learned by MECHANIC while varying batch size and model size.

time. This could easily have been a potential failure mode of any kind of automatic learning rate tuner as such strategies might "explore" a high learning rate at the beginning of the optimization procedure. Fortunately, we observed that this inductive bias typically baked at finetuning time is still maintained when using MECHANIC.

### 4.2 Image Classification

In this Section, we present results on popular Image Classification tasks. Apart from training from scratch, we also perform transfer learning experiments where we pre-train on the JFT-300M [40] dataset and finetune on ImageNet, Cifar-10 and Cifar-100 datasets. We follow the exact setting employed in [41] for both pre-training and finetuning.

As shown in Table 2, MECHANIC is quite competitive across the board and produces results either very close to the baseline or better. Since MECHANIC optimizes for the train loss, in general, we observe that it results in better **test** performance on tasks with large amounts of data where the model is unable to overfit to the train set. For instance, we see that MECHANIC beats the baseline substantially when pre-training ViT models on JFT-300M, whereas it lags slightly behind on smaller datasets like ImageNet-1k or CIFAR-10/100. Even though we fix $\lambda$ to 0.01 as default for all our reported experiments, we find that for small datasets like CIFAR-10, increasing it led to better test performance.

### 4.3 Comparison with D-adaptation

Recently, [16] introduced the D-adaptation algorithm, with the same goal of learning the correct scale $s$ for SGD and Adam base optimizers. D-adaptation showed impressive empirical results on a range of popular deep learning tasks, so we compare MECHANIC with D-adaptation on a selection of tasks that D-adaptation worked well on, using code provided by the authors. Hyper-parameter settings were kept the same to ensure a fair comparison. In contrast to D-adaptation, MECHANIC does not require modification for different base optimizers and, as shown in Figure 2, it remains quite

| Model | Size | Pre Opt | Pre Acc | Optimizer | I1K | Cifar100 | Cifar10 |
|-------|------|---------|---------|-----------|-----|----------|---------|
| | | | | CNN from scratch on CIFAR datasets | | | |
| ResNet-18 | 11M | - | - | Mom | - | **77.6** | **95.4** |
| | | - | - | $\mathcal{M}$-Mom | - | 75.3 | 94.1 |
| | | - | - | $\mathcal{M}$-Mom ($\lambda = 0.1$) | - | 76.6 | 95.3 |
| WRN-40-10 | 56M | - | - | Mom | - | **79.9** | - |
| | | - | - | $\mathcal{M}$-Mom | - | 79.6 | - |
| | | | | Pre-train on JFT-300M | | | |
| ViT-B/16 | 86M | AdamW | 48.5 | Mom | 84.7 | **91.9** | 99.1 |
| | | | | $\mathcal{M}$-Mom | **84.7** | 90.7 | 99.1 |
| | | $\mathcal{M}$-AdamW | **49.9** | Mom | 84.2 | 91.5 | 99.1 |
| | | | | $\mathcal{M}$-Mom | 84.1 | 90.3 | **99.1** |
| ViT-B/16 | 86M | Lion | 47.0 | Mom | **85.3** | 92.1 | 99.2 |
| | | | | $\mathcal{M}$-Mom | 85.2 | 91.0 | 99.2 |
| | | $\mathcal{M}$-Lion | **49.6** | Mom | 84.7 | **92.3** | **99.2** |
| | | | | $\mathcal{M}$-Mom | 84.6 | 90.9 | 99.1 |
| ViT-L/16 | 307M | AdamW | 54.4 | Mom | **86.7** | **93.9** | 99.5 |
| | | | | $\mathcal{M}$-Mom | 86.6 | 92.7 | **99.5** |
| | | $\mathcal{M}$-AdamW | **54.4** | Mom | 86.3 | 93.4 | 99.4 |
| | | | | $\mathcal{M}$-Mom | 86.0 | 92.0 | 99.3 |
| ViT-L/16 | 307M | Lion | 52.0 | Mom | 86.7 | 93.8 | 99.4 |
| | | | | $\mathcal{M}$-Mom | 86.7 | 93.0 | 99.4 |
| | | $\mathcal{M}$-Lion | **55.4** | Mom | 87.2 | **94.0** | 99.4 |
| | | | | $\mathcal{M}$-Mom | **87.2** | 93.4 | **99.4** |

Table 2: Comparing MECHANIC on vision models. All fine-tuning results are averaged over 3 independent runs with different seeds.

competitive on small datasets like CIFAR-10/100 while outperforming both a manually tuned baseline and D-adaptation on bigger tasks like IWSLT14 and language modeling on BookWiki dataset. We present additional results in Appendix C.3, including a comparison on a suite of 12 logistic regression problems.

## 5 Conclusion

MECHANIC is a new technique for scaling the updates of any base optimization algorithm. Our approach provides a practical implementation of recent developments in optimization theory, and is able to match the performance of tuned baselines on large-scale machine learning tasks. This work suggests several natural future directions. First, is there a theoretical motivation for our weight-decay term? Next, is it possible to leverage similar techniques to learn a *per-layer* scale factor? Such a capacity would not significantly increase computation cost, but by allowing more degrees of freedom may yield a method that significantly outperforms baselines since it is infeasible to manually tune a scale factor for every layer.

### Acknowledgments

Ashok Cutkosky acknowledges funding support from NSF grant CCF-2211718, an Amazon research award, and Google.

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

# A  Limitations

In this paper, we introduce a technique for automatically learning the right scale of the learning rate called MECHANIC and evaluate it on a broad range of practical deep learning problems and settings. We find that, depending on the problem, MECHANIC can be quite effective and even surpass performance of manual tuning of learning rates at a fraction of the cost. We also find that in addition to training from scratch, MECHANIC also works for finetuning.

While the initial set of results are encouraging, many challenges remain. Firstly, we found that MECHANIC does not seem to work well with dropout [42]. While MECHANIC is effective against noise from sampling, we believe there may be a more fundamental reason why dropout does not work well with MECHANIC. Second, MECHANIC re-purposes the gradient coming from the *train set* for learning the learning rate, which means it optimizes for train loss. This is different from manual tuning of learning rates where researchers tune it based on performance on the *validation* set. A principled way to handle this discrepancy is also in interesting avenue of future research.

# B  Additional Ablations

## B.1  Setting BASE learning rate with Mechanic

Even though we recommend setting peak learning rate of Base optimizer to 1.0, to make sure that Mechanic truly is insenstive to the peak learning rate and robust to the choice of this hyperparameter, we conduct a study where we vary peak learning rate of AdamW on BERT-B masked language modeling task. As shown in Table 3, Mechanic is largely robust against choice of peak LR set for AdamW on BERT-B MLM task.

| Peak LR of BASE with Mechanic | MLM Acc |
|---|---|
| 1e-2 | 71.7 |
| 1e-1 | 71.6 |
| 1e0 | 71.6 |
| 1e1 | 71.4 |
| 1e2 | 71.6 |

Table 3: MECHANIC is largely robust against choice of peak LR set for AdamW on BERT-B MLM task.

## B.2  Robustness to $s_{init}$

| $s_{init}$ | 1e-8 | 1e-7 | 1e-6 | 1e-5 | 1e-4 |
|---|---|---|---|---|---|
| Accuracy | 49.8 | 49.8 | 49.9 | 49.7 | 49.6 |

Table 4: Accuracy on JFT-300M using model ViT-B/16 and optimizer $\mathcal{M}$-AdamW as a function $s_{init}$. MECHANIC is robust to varying this parameter.

## B.3  Ablation of $\lambda$

| $\lambda$ | 0 | 1e-3 | 1e-2 | 1e-1 | 1e0 |
|---|---|---|---|---|---|
| Accuracy | 49.7 | 49.8 | 49.9 | 49.7 | Diverged |

Table 5: Accuracy on JFT-300M using model ViT-B/16 and optimizer $\mathcal{M}$-AdamW as a function $\lambda$. We have observed that while $\lambda$ is helpful in stabilizing MECHANIC on some problems, as long as it is set to a reasonable small value it does not affect performance by a lot.

## B.4 Number of $\beta$ values

| Number of $\beta$ values ($n$) | 2 | 4 | 6 | 8 |
|---|---|---|---|---|
| Accuracy | 48.9 | 49.5 | 49.9 | 49.6 |

Table 6: Accuracy on JFT-300M using model ViT-B/16 and optimizer $\mathcal{M}$-AdamW as a function $n$, the number of $\beta$ values. The $\beta$ values are set as $1 - 0.1^i$ for $i$ from 1 to $n$. This is the most sensitive parameter in MECHANIC. For smaller $n$ we see some significant performance degradation, while for larger $n$ we see milder degradation. Theory suggests that larger $n$ should result in a degradation that is roughly logarithmic in $n$.

## C  Additional Experimental Details

### C.1  Hyperparams for BERT

| Model | Aug | Optimizer | $\beta_1$ | $\beta_2$ | lr sweep | best lr | Weight decay |
|---|---|---|---|---|---|---|---|
| BERT-B | | AdamW | 0.9 | 0.999 | [5e-4, 1e-3, 2e-3, 5e-3, 1e-2] | 5e-3 | |
| BERT-L | | AdamW | 0.9 | 0.999 | [5e-4, 1e-3, 2e-3, 5e-3, 1e-2] | 1e-3 | |
| BERT-B/L | | $\mathcal{M}$-AdamW | 0.9 | 0.999 | | | 0.01 |
| BERT-B | | Lion | 0.9 | 0.99 | [5e-5, 1e-4, 2e-4, 5e-4, 1e-3] | 5e-4 | |
| BERT-L | | Lion | 0.9 | 0.99 | [5e-5, 1e-4, 2e-4, 5e-4, 1e-3] | 2e-4 | |
| BERT-B/L | | $\mathcal{M}$-Lion | 0.9 | 0.99 | | | 0.1 |

Table 7: Critical hyperparameters we used for BERT pre-training. For the baselines we grid searched the learning rate as shown in the table. Batch size 2k, trained for 150k steps on original Wikibooks dataset w/o NSP loss (similar to Roberta). We found that a small amount of weight decay makes MECHANIC slightly more effective.

| Model | | AdamW LR | | | | | MECHANIC-AdamW |
|---|---|---|---|---|---|---|---|
| BERT-B | 71.1 | 71.5 | 71.5 | 71.5 | 71.3 | | 71.7 |
| BERT-L | 75.0 | 75.4 | 75.4 | 74.6 | 74.4 | | 75.3 |
| | | Lion LR | | | | | MECHANIC-Lion |
| BERT-B | 70.8 | 70.8 | 71.1 | 71.8 | 71.4 | | 72.0 |
| BERT-L | 75.1 | 75.6 | 75.7 | 74.7 | Diverged | | 75.5 |

Table 8: As detailed in Table 7, we performed a grid search of learning rate values for each base optimizer. Here, we present the resulting accuracy values.

| Hyperparam | Optimizer | Without MECHANIC pre-training | With MECHANIC pre-training |
|---|---|---|---|
| Learning Rate | Adam | [5e-5, 1e-4, 2e-4, 3e-4, 5e-4] | [1e-5, 2e-5, 3e-5, 5e-5, 1e-4] |
| Learning Rate | Lion | [5e-6, 1e-5, 2e-5, 3e-5, 5e-5] | [1e-6, 2e-6, 3e-6, 5e-6, 1e-5] |
| Batch Size | | [16, 32] | [16, 32] |
| Weight Decay | | 0 | 0 |
| Max Epochs | | 10 | 10 |
| Learning Rate Decay | | Linear | Linear |
| Warmup Ratio | | 0.06 | 0.06 |
| Dropout | | 0.1 | 0.1 |
| Attention Dropout | | 0.1 | 0.1 |

Table 9: BERT GLUE finetuning hparams with AdamW.

| Hyperparam | Value |
|---|---|
| Batch Size | [16, 32] |
| Weight Decay | 0 |
| Max Epochs | 10 |
| Learning Rate Decay | Linear |
| Warmup Ratio | 0.06 |
| Dropout | 0.0 |
| Attention Dropout | 0.1 |

Table 10: BERT GLUE finetuning hparams when using mechanic at finetuning time. We found a limitations of MECHANIC that it does not perform well in combination to dropout, so we set dropout rate to 0 for these experiments.

## C.2 Hyperparams for Image Classification

| Model | Aug | Optimizer | $\beta_1$ | $\beta_2$ | lr | Weight decay | Num epochs |
|---|---|---|---|---|---|---|---|
| ViT-B/16 | | AdamW | 0.9 | 0.999 | 8e-4 | 0.1 | 7 |
| ViT-B/16 | | $\mathcal{M}$-AdamW | 0.9 | 0.999 | | 0.1 | 7 |
| ViT-L/16 | | AdamW | 0.9 | 0.999 | 4e-4 | 0.1 | 7 |
| ViT-L/16 | | $\mathcal{M}$-AdamW | 0.9 | 0.999 | | 0.1 | 7 |
| ViT-B/16 | | Lion | 0.9 | 0.99 | 1e-4 | 0.3 | 7 |
| ViT-B/16 | | $\mathcal{M}$-AdamW | 0.9 | 0.99 | | 0.3 | 7 |
| ViT-L/16 | | Lion | 0.9 | 0.99 | 1e-4 | 0.3 | 7 |
| ViT-L/16 | | $\mathcal{M}$-AdamW | 0.9 | 0.99 | | 0.3 | 7 |

Table 11: Critical hyperparameters we used for all the experiments, most of them directly repurposed from [41]. For each baseline we repurposed a well-tuned base learning from previous work [41, 43]. Trained on JFT-300M with batch size 4k with LR cosine decay schedule.

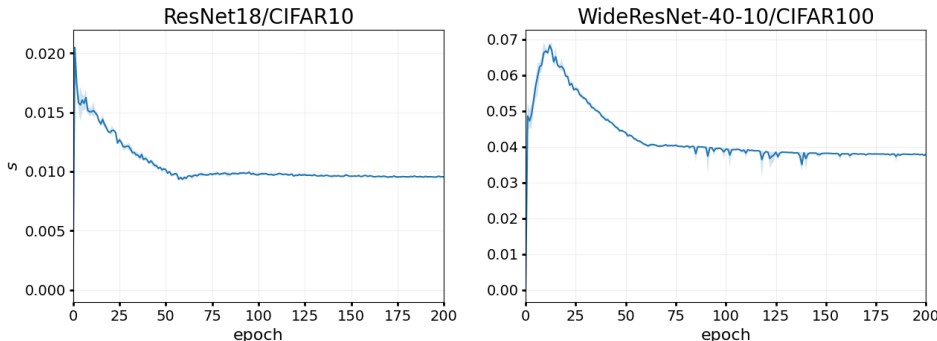

Figure 3: Scaling values $s$ learned by MECHANIC during ResNet18 training on CIFAR10 and WideResNet training on CIFAR100. Shaded area represents max/min value over 3 runs. Dark line is average.

| Hyperparam | ImageNet | CIfar100 | Cifar10 |
|---|---|---|---|
| Learning rate sweep | $\{0.003, 0.01, 00.03, 0.06\}$ | $\{0.001, 0.003, 0.01, 00.03\}$ | $\{0.001, 0.003, 0.01, 00.03\}$ |
| Batch size | 512 | 512 | 512 |
| Weight decay | 0 | 0 | 0 |
| Num steps | 20k | 10k | 10k |
| Warmup steps | 500 | 500 | 500 |
| Learning rate decay | Cosine | Cosine | Cosine |
| Dropout | 0.0 | 0.0 | 0.0 |
| Clipping norm | 1.0 | 1.0 | 1.0 |

Table 12: We directly use ViT finetuning hyperparams recommended by [41]. For MECHANIC we also use same hyperparameters, omitting just the learning rate sweep, since we don't need it now. We use finetuning resolution of 384.

| Hyperparam | Value |
|---|---|
| weight decay | $[0.001, 0.0005, 0.0001]$ |
| lr | $[0.3, 0.1, 0.03]$ |
| SGD Momentum $\beta$ | 0.9 |
| batch size | 128 |
| num epochs | 200 |
| schedule | Step decay by 0.2 at 60, 120, 160 epochs |
| augmentations | Random Crop, Random Horizontal Flip |
| Gradient clip by global norm | 1.0 |
| $\lambda$ | Kept at default 0.01 |

Table 13: Hyperparameters for tuning ResNet18 on CIFAR10 and WideResNet on CIFAR100

### C.3 Hyperparams and additional results on comparisons with D-adaptation

## D Theoretical Analysis

Here we provide the theoretically tractable version of TUNER as well as its analysis.

### D.1 Algorithmic Simplifications

To simplify the implementation of MECHANIC, we replaced all of the red text in Algorithm 2 with a single line $r_t \leftarrow \max(0, r_t)$ right after the definition of $r_t$. This is motivated by two ideas.

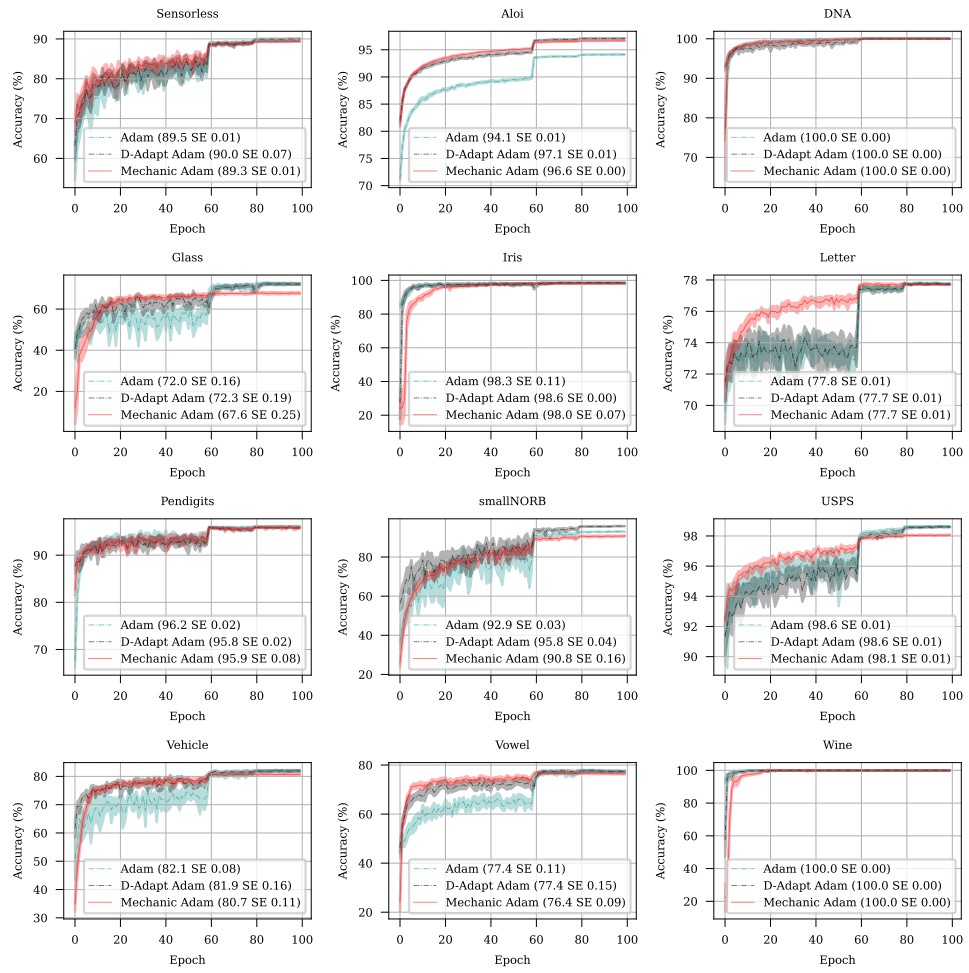

Figure 4: Comparing MECHANIC with D-adaptation and manually tuned learning rates on a suite of convex tasks.

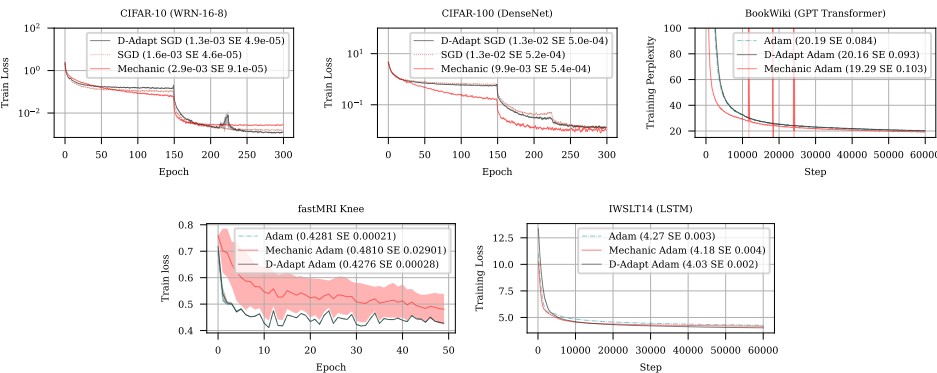

Figure 5: Complementary train set results of MECHANIC with D-adaptation and manually tuned learning rates on vision and language tasks.

---

**Algorithm 2** TUNER(theoretically tractable version

---

1: **Input:** $\beta \in [0, 1]$, $W_0$: initial "wealth".
2: $v_0 \leftarrow 0 \in \mathbb{R}$, $r_0 \leftarrow 0 \in \mathbb{R}$, $\mathbf{q}_t \leftarrow 0 \in \mathbb{R}$, $m_0 \leftarrow 0 \in \mathbb{R}$, $s_1 \leftarrow 0 \in \mathbb{R}$
3: **for** $t = 1 \ldots T$ **do**
4:     Output $s_t$.
5:     Receive $h_t$.
6:     $m_t \leftarrow \max(\beta m_{t-1}, h_t)$
7:     $\hat{h}_t = \text{clip}(h_t, -m_{t-1}, m_{t-1})$ // Red text are steps that we omit in practice (see Algorithm 1)
8:     $\mathbf{q}_t \leftarrow \beta \mathbf{q}_{t-1} + \hat{h}_t$
9:     $r_t \leftarrow \beta r_{t-1} - s_{t-1} \hat{h}_t$
10:     $W_t \leftarrow$ $W_0$ $+ r_t$ // Blue text is changed in practice (see Algorithm 1)
11:     $v_t \leftarrow \beta^2 v_{t-1} + \hat{h}_t^2$
12:     $s_{t+1} \leftarrow \dfrac{W_t}{\sqrt{4m_t^2 + v_t} + \epsilon} \cdot \text{clip}(\mathbf{q}_t / \sqrt{4m_t^2 + v_t}, 0, 1)$
13: **end for**

---

First, we conjecture that the clip operation using $\mathbf{q}_t / \sqrt{v_t}$ may even be unnecessary in theory[2]: we observed no change from removing this operation in practice, and observe that the update has an intuitive interpretation via the Taylor expansion discussed in Section 3.

Second, the clip operation on $h_t$ using $m_{t-1}$ is essentially designed to prevent the wealth $W_t$ from becoming negative or zero using the gradient truncation technique employed by [44, 45, 12]. While less consistent with known theory, we found it simpler to ensure the wealth $W_t$ does become negative simply by clipping $r_t$ directly (we did not clip $W_t$ to be nonnegative as $W_t = 0$ would cause the algorithm to output $s_t = 0$ for all future iterations). We found these changes simplified the algorithm while having no noticeable effect on the performance. Although these deviations technically do not come with guarantees, the accomplish similar intuitive goals and so we expected (and observed) that they simplified the implementation while not damaging performance.

### D.2   Eliminating $W_0$ in favor of $s_{init}$

While TUNER makes use of the "initial wealth" value $W_0$, MECHANIC instead adopts a varying value for $W_0$ proportional to $s_{init} \cdot m_t$. This makes the first $s$ value proposed by MECHANIC equal to $s_{init}$, which is more intuitive to set than $W_0$. The exponential growth in $s$ allows us to set $s_{init}$ to a very small value of $10^{-8}$. It also makes the values for $s$ "scale-free" in the sense that rescaling the updates $\mathbf{u}_t$ by any constant will have no effect on the resulting $s_t$.

### D.3   Regret Bound

**Theorem 2.** *With $\beta = 1$, Algorithm 2 guarantees for all $\mathring{s} \geq 0$:*

$$\sum_{t=1}^{T} h_t(s_t - \mathring{s}) \leq W_0 + (\mathring{s} + \max_t s_t) \cdot m_T + O\left(\mathring{s} \cdot \log(T\mathring{s}m_T/m_1 W_0)\sqrt{\sum_{t=1}^{T} h_t^2}\right).$$

*Proof.* First, we employ an argument developed by [44]:

$$\sum_{t=1}^{T} h_t(s_t - \mathring{s}) \leq \sum_{t=1}^{T} \hat{h}_t(s_t - \mathring{s}) + \sum_{t=1}^{T} |\hat{h}_t - h_t|(|s_t| + |\mathring{s}|)$$

$$\leq \sum_{t=1}^{T} \hat{h}_t(s_t - \mathring{s}) + (\max_t |s_t| + |\mathring{s}|) \sum_{t=1}^{T} |\hat{h}_t - h_t|$$

$$= \sum_{t=1}^{T} \hat{h}_t(s_t - \mathring{s}) + m_T(\max_t |s_t| + |\mathring{s}|).$$

---

[2]Removing the clip in theory may requiring some additional non-worst-case assumption.

So, in the following, it suffices to bound $\sum_{t=1}^{T} \hat{h}_t(s_t - \mathring{s})$. This is helpful because we will be able to use the bound $|\hat{h}_t| \leq m_{t-1}$, and $m_{t-1}$ is known *before* $\hat{h}_t$ is revealed.

As is typical in the analysis of parameter-free algorithms, the proof proceeds by lower-bounding the wealth. Define a function $a(x)$ piecewise by:

$$a(x) = \begin{cases} 0 & x \leq 0 \\ x^2/2 & x \in [0,1] \\ x - 1/2 & x \geq 1 \end{cases}$$

Notice that $a(x)$ is differentiable, monotonically increasing and 1-Lipschitz. We are going to roughly show that $W_t \geq \Omega(\exp(a(-\sum_{k=1}^{t} \hat{h}_k/\sqrt{v_t})))$, after which the regret bound will follow from the wealth-regret duality [10].

The key technical inequality in the proof is the following: for any $A$, $B$, $m$ with $B \geq 4m^2$, and any $|x| \leq m$, we have:

$$a\left(\frac{-A}{\sqrt{B}}\right) - \frac{x}{\sqrt{B}}\mathrm{clip}\left(\frac{-A}{\sqrt{B}}, 0, 1\right) \geq a\left(\frac{-A-x}{\sqrt{B+x^2}}\right) - \frac{x^2}{B}. \tag{6}$$

Once (6) is established, we proceed as follows: defining $c_t = \frac{\mathrm{clip}(\sum_{k=1}^{t-1} \hat{h}_k/\sqrt{4m_{t-1}^2+v_{t-1}},0,1)}{\sqrt{v_t}}$, we have:

$$\log(W_t) = \log(W_{t-1}) + \log(1 - \hat{h}_t c_t)$$
$$\geq \log(W_{t-1}) - \hat{h}_t c_t - \hat{h}_t^2 c_t^2,$$

where we have used $c_t \leq 1/2$ and the identity $\log(1 - x) \geq -x - x^2$ for $x \leq 1/2$ (which applies since $\hat{h}_t \leq m_{t-1}$ by definition). Now, set $A = \sum_{k=1}^{t-1} \hat{h}_k$ and $B = 4m_{t-1}^2 + v_{t-1}$ and $x = \hat{h}_t$ in (6), we see that:

$$\log(W_t) - \log(W_{t-1}) \geq -\frac{x}{\sqrt{B}}\mathrm{clip}\left(\frac{-A}{\sqrt{B}}, 0, 1\right) - \frac{\hat{h}_t^2}{4m_{t-1}^2 + \mathbf{v}_{t-1}}$$

$$\geq a\left(\frac{-A-x}{\sqrt{B+x^2}}\right) - a\left(\frac{-A}{\sqrt{B}}\right) - \frac{x^2}{B} - \frac{\hat{h}_t^2}{4m_{t-1}^2 + \mathbf{v}_{t-1}}$$

$$\geq a\left(\frac{-\sum_{k=1}^{t} \hat{h}_k}{\sqrt{4m_{t-1}^2 + v_t}}\right) - a\left(\frac{-\sum_{k=1}^{t-1} \hat{h}_k}{\sqrt{4m_{t-1}^2 + v_{t-1}}}\right) - \frac{2\hat{h}_t^2}{4m_{t-1}^2 + v_{t-1}}$$

$$\geq a\left(\frac{-\sum_{k=1}^{t} \hat{h}_k}{\sqrt{4m_t^2 + v_t}}\right) - a\left(\frac{-\sum_{k=1}^{t-1} \hat{h}_k}{\sqrt{4m_{t-1}^2 + v_{t-1}}}\right) - \frac{2\hat{h}_t^2}{4m_{t-1}^2 + v_{t-1}}.$$

Thus by telescoping the sum, we have:

$$\log(W_T) \geq \log(W_0) + a\left(\frac{-\sum_{k=1}^{T} \hat{h}_k}{\sqrt{v_T}}\right) - \sum_{t=1}^{T} \frac{2\hat{h}_t^2}{4m_{t-1}^2 + v_{t-1}}.$$

Now, observe that $\frac{2\hat{h}_t^2}{4m_{t-1}^2 + v_{t-1}} \leq \frac{2\hat{h}_t^2}{v_t} \leq 2(\log(v_t) - \log(v_{t-1}))$, so we have $\sum_{t=1}^{T} \frac{\hat{h}_t^2}{v_{t-1}} \leq 2\log(Tm_T/m_1)$ so that overall:

$$W_T \geq \frac{W_0 m_1}{T^2 m_T}\exp\left[a\left(\frac{-\sum_{k=1}^{T} \hat{h}_k}{\sqrt{v_T}}\right)\right].$$

Thus, if we define $p(H) = \frac{W_0 m_1}{T^2 m_T} \exp\left[a\left(\frac{H}{\sqrt{v_T}}\right)\right]$, we have $W_T \geq p(-\sum_{k=1}^T \hat{h}_k)$. Now, we employ the reward-regret duality:

$$\sum_{t=1}^T \hat{h}_t(s_t - \mathring{s}) = s_{init} \cdot m + \mathring{s}\sum_{t=1}^T(-\hat{h}_t) - W_T$$

$$\leq W_0 + \sup_G \mathring{s} \cdot G - p(G)$$

$$= W_0 + p^\star(\mathring{s})$$

$$\leq W_0 + O(s\log(sT/s_{init})\sqrt{v_T}).$$

Where $p^\star$ is the Fenchel conjugate of $p$ and the evaluation of the conjugate follows from direct calculation (see, e.g. [10, 28, 46]).

Thus, to prove the theorem we need only show (6). This is established via casework in a manner similar to [46].

**Case 1.** $\frac{-A}{\sqrt{B}} \leq 0$: In this case, the statement is equivalent to: $\frac{x^2}{B} \geq a\left(\frac{-A-x}{\sqrt{B+x^2}}\right)$. Note that since $\frac{-A}{\sqrt{B}} \leq 0$, we have $A \geq 0$. Therefore:

$$\frac{-A-x}{\sqrt{B+x^2}} = \frac{-A}{\sqrt{B+x^2}} - \frac{x}{\sqrt{B+x^2}}$$

$$\leq -\frac{x}{\sqrt{B+x^2}}.$$

Further, we clearly have $-\frac{x}{\sqrt{B+x^2}} \leq 1$ so that:

$$a\left(\frac{-A-x}{\sqrt{B+x^2}}\right) \leq a\left(-\frac{x}{\sqrt{B+x^2}}\right) = \frac{x^2}{2(B+x^2)} \leq \frac{x^2}{B}.$$

So, in the following we assume $\frac{-A}{\sqrt{B}} \geq 0$.

**Case 2.** $\frac{-A-x}{\sqrt{B+x^2}} \leq 0$: In this case, it suffices to show $\frac{-x}{\sqrt{B}}\text{clip}\left(\frac{-A}{\sqrt{B}}, 0, 1\right) \geq -\frac{x^2}{B}$. The case assumption implies $m^2 \geq x \geq -A \geq 0$. Therefore, since $B \geq 4m^2$, $\text{clip}\left(\frac{-A}{\sqrt{B}}, 0, 1\right) = \frac{-A}{\sqrt{B}}$ so that $\frac{-x}{\sqrt{B}}\text{clip}\left(\frac{-A}{\sqrt{B}}, 0, 1\right) = \frac{xA}{B} \geq \frac{-x^2}{B}$ as desired.

So, in the following we now further assume $\frac{-A-x}{\sqrt{B+x^2}} \geq 0$.

**Case 3.** $\frac{-A}{\sqrt{B}} \in [0, 1]$: We have $a\left(\frac{-A}{\sqrt{B}}\right) = \frac{A^2}{2B}$, and also since $a(z) \leq \frac{1}{2}z^2$ for all $z$, $a\left(\frac{-A-x}{\sqrt{B+x^2}}\right) \leq \frac{(A+x)^2}{2(B+x^2)}$. Thus, it suffices to show that $\frac{A^2}{2B} + \frac{xA}{B} \geq \frac{(A+x)^2}{2(B+x^2)} - \frac{x^2}{B}$, but this is equivalent to $\frac{(A+x)^2}{2B} \geq \frac{(A+x)^2}{2(B+x^2)} - \frac{x^2}{2B}$, which clearly holds.

**Case 4:** $\frac{-A}{\sqrt{B}} \geq 1$ **and** $\frac{-A-x}{\sqrt{B+x^2}} \geq 1$: In this case it suffices to show $\frac{-A}{\sqrt{B}} - \frac{x}{\sqrt{B}} \geq \frac{-A-x}{\sqrt{B+x^2}} - \frac{x^2}{B}$. To see this, we have:

$$\frac{-A-x}{\sqrt{B+x^2}} = \frac{-A}{\sqrt{B+x^2}} - \frac{x}{\sqrt{B+x^2}}$$

$$\leq \frac{-A}{\sqrt{B}} - \frac{x}{\sqrt{B}} + x\left(\frac{1}{\sqrt{B}} - \frac{1}{\sqrt{B+x^2}}\right)$$

$$\leq \frac{-A}{\sqrt{B}} - \frac{x}{\sqrt{B}} + \frac{2x^3}{3B^{3/2}}$$

$$\leq \frac{-A}{\sqrt{B}} - \frac{x}{\sqrt{B}} + \frac{x^2}{B},$$

where in the second-to-last line we have used the fact that $h \mapsto \frac{1}{\sqrt{B+h}}$ is a convex in $h$, and in the last line we have used $\sqrt{B} \geq m \geq x$.

**Case 5:** $\frac{-A}{\sqrt{B}} \geq 1$ **and** $\frac{-A-x}{\sqrt{B+x^2}} \in [0,1)$**:** In this case we need to show $\frac{-A}{\sqrt{B}} - \frac{1}{2} \geq \frac{(A+x)^2}{2(B+x^2)} - \frac{x^2}{B} + \frac{x}{\sqrt{B}}$. To see this, we first observe that since $\frac{-A-x}{\sqrt{B+x^2}} \in [0,1)$, we have

$$A^2 + 2Ax + x^2 \leq B + x^2$$
$$A^2 + 2Ax \leq B.$$

Thus, by quadratic formula, $A \geq -x - \sqrt{x^2+B}$, so that we have $A \in [-x - \sqrt{x^2+B}, -\sqrt{B}]$.

Next, our target identity can be rearranged into an equivalent form as follows:

$$\frac{-A}{\sqrt{B}} - \frac{1}{2} \geq \frac{(A+x)^2}{2(B+x^2)} - \frac{x^2}{B} + \frac{x}{\sqrt{B}}$$

$$0 \geq \frac{(A+x)^2}{2(B+x^2)} + \frac{A}{\sqrt{B}} + \frac{1}{2} - \frac{x^2}{B} + \frac{x}{\sqrt{B}},$$

so that it suffices to show the second line above. Notice that the RHS of this expression is convex in $A$ and so is maximized at the boundary of the range $[-x - \sqrt{x^2+B}, -\sqrt{B}]$. When $A = -\sqrt{B}$ we have:

$$\frac{(A+x)^2}{2(B+x^2)} + \frac{A}{\sqrt{B}} + \frac{1}{2} - \frac{x^2}{B} + \frac{x}{\sqrt{B}} \leq \frac{(A+x)^2}{2B} + \frac{A}{\sqrt{B}} + \frac{1}{2} - \frac{x^2}{B} + \frac{x}{\sqrt{B}} = -\frac{x^2}{2B} \leq 0.$$

Alternatively, when $A = -x - \sqrt{x^2+B}$, we have

$$\frac{(A+x)^2}{2(B+x^2)} + \frac{A}{\sqrt{B}} + \frac{1}{2} - \frac{x^2}{B} + \frac{x}{\sqrt{B}} = 1 - \frac{\sqrt{x^2+B}}{\sqrt{B}} - \frac{x^2}{B}$$
$$\leq 0.$$

This establishes the claimed inequality (6) and completes the proof.

$\square$

