# OpenReview forum: "Mechanic: A Learning Rate Tuner"
_NeurIPS.cc/2023/Conference — NeurIPS 2023 poster_

### Official Review · Reviewer_DR7W · 2023-07-07

**Soundness:** 3 good
**Presentation:** 3 good
**Contribution:** 4 excellent
**Rating:** 7
**Confidence:** 3

**Summary:**

This paper focuses on a method (MECHANIC) for tuning the learning rate of any given base optimizer. This is compatible with any given base optimizer along with a learning rate schedule*. Let u_1, u_2,..u_t be the steps of the base optimizer up until now, then MECHANIC chooses a s_t such that the neural network weights are set to w_{init} + s_t (\sum_i u_i). They do this by leveraging a similar scheme developed for online convex optimization. They implement this method with some minor changes to simplify the implementations and improve the convergence speed (by using momentum and a variant of weight decay).

The authors show that in experiments on masked language modeling and image classification, using MECHANIC leads to faster optimization (although it does sometimes lead to worse generalization).

*One thing to note is that this is not a replacement for learning rate schedulers. In the CIFAR10 plots (Fig 2), there is still a need for learning rate decay. Rather, it seems to help over the tuned baselines for any given learning rate schedule.

Typos:
Line 32: The end quote is flipped.
Line 17 in algorithm: s_{t+1,n} should be s_{t+1,i}.
Line 113: Text is missing.
Table 3, last line: MAdamW should be M-LION.

**Strengths:**

Improving over tuned baselines is an important contribution. The authors do so for large scale deep learning datasets such as BERT and JFT-300m. It is a bonus that the authors are able to do so by using a theoretically motivated algorithm.


**Weaknesses:**

The results would have been more impressive if the authors had used more recent baselines such as Mosaic ML’s BERT codebase.

**Questions:**

1. Could the authors provide some intuition about the fact that s_t is multiplied to the sum of the steps taken by the network till now rather than the next step (as is done in learning rate schedulers).
2. It is surprising that the learned value of s (Figure 1 and 3) is usually  << .1? For comparison against tuned learning rates I would have expected s to be not significantly smaller/larger than 1. Does this suggest that the learning rates being compared to could have been reduced by a factor of 10 without significant effects? Could the authors provide some intuition about the learnt values of s?
3. The main body says that lamba is always set to .01 while Table 2 says that lambda=.1 in one of the cases. Was lambda tuned for MECHANIC? If so, was weight decay tuned for the base optimizers?
4. I may have missed the justification referred to in “This seems like a significant issue to disregard, but we will provide mathematical justification presently.”. Could the authors point where this is? Also, I think the first line of the proof of 3.1 is assuming this i.e. that g_t is independent of s_t.
5. Line 100: What is “s”? The previous lines only had s^{o}.


**Limitations:**

Yes.

---

> ### Author Rebuttal · Authors · 2023-08-10
>
> Thank you for your positive comments and work reviewing our paper! We have included answers to your questions below, and we’d be happy to elaborate further.
>
> Intuition for scaling by the sum of the steps: At a high level, the sum of the steps is a more “stable” value because it changes slowly. Individual steps, in contrast, change direction very quickly and so make it difficult to stabilize on the correct learning rate. In a technical sense, this manifests in analyses of algorithms that attempt to learn the scale factor applied to each step often needing to solve optimization problems with high variance or Lipschitz constants and so are unable to ensure quick convergence.
>
> The scale factor is applied to the base learning updates with no learning rate (or, if you prefer, a learning rate of 1.0). This is why s is smaller than 1.
>
> We did tune lambda for mechanic, as well as weight decay for the base optimizers. We found that without lambda, mechanic still performed well, but not quite as well (especially on smaller datasets). Please see the response to reviewer mCwi and the ablation plots included in the general response for more detail.
>
> The justification is provided in Theorem 1, for which the regret of the base algorithm is measured with gradients evaluated at rescaled iterates rather than the original base iterates. We are not sure what you mean by the first line of the proof of 3.1: what is 3.1? We don’t think we ever assumed any kind of independence: in fact, our proofs do not assume any kind of stochastic structure at all and hold even for completely adversarial gradient or loss sequences, which is typical in online convex optimization. Specifically, our analysis has the following overall structure:
>
> First, by convexity one has $\mathbb{E}\left[\sum_{t=1}^T F(x_t) -F(\mathring{x}) \right]\le \mathbb{E}\left[\sum_{t=1}^T  \langle g_t, x_t - \mathring{x}\rangle\right]$.
>
> Second, we prove that for *any* sequence of vectors $g_1,\dots,g_t$, the quantity $\sum_{t=1}^T  \langle g_t, x_t - \mathring{x}\rangle$ is small. In particular, this part of the analysis actually does not need the $g_t$ to be gradients of anything - they can be arbitrarily and even adversarially generated. Please check out the response for reviewer 7tau for more intuition behind the analysis.
>
> Line 100:  $s$ is the scale value used in the base implementation of SGD, as described in equations (1) and (2). $\mathring{s}$ is the “correct” scale value that we should have used.

---

> > ### Comment · Reviewer_DR7W · 2023-08-13
> > **Reply to Authors**
> >
> > >3.1
> >
> > I meant theorem 1.
> >
> >
> > > "The justification is provided in Theorem 1, for which the regret of the base algorithm is measured with gradients evaluated at rescaled iterates rather than the original base iterates. "
> >
> > > "Second, we prove that for any sequence of vectors"
> >
> > Does this is point to a gap between the theorem and the application? or do you mean the theorem is more general then needed?
> >
> > > "or, if you prefer, a learning rate of 1.0"
> >
> > I see, so it starts with a higher learning rate than the base algorithm and then learns to decrease the learning rate in a near optimal way. Then why is the schedule needed at all i.e. why does MECHANIC not decrease the learning rate automatically?

---

> > > ### Author Response · Authors · 2023-08-14
> > >
> > > "Does this is point to a gap between the theorem and the application? or do you mean the theorem is more general then needed?"
> > >
> > > The theorem is more general than needed. It holds for arbitrary sequences of vectors, when in reality we only require it to hold for sequence of vectors that are stochastic gradients of a loss. This is actually pretty common in the analysis of optimization algorithms, although it seems very unintuitive at first blush - you'd think that the supposedly harder setting makes you lose something, but it turns out that the achieved convergence results cannot be improved even if you do assume stochastic structure. Basically, it turns out that the "worst case" sequence of vectors usually actually tends to be a simple stochastic sequence and so you don't gain much by making the stochastic assumption. Does this also clear up the confusion about independence in the first line of the proof?
> > >
> > > "I see, so it starts with a higher learning rate than the base algorithm and then learns to decrease the learning rate in a near optimal way"
> > >
> > > Not quite: it starts with a very small learning rate scale of $s_{init}$ (say 1e-8), and then very very quickly increases the value to reach the "correct" scale - so quickly that it appear near immediate in the plots. In practice, it tends to actually overshoot a bit and then come back down.
> > >
> > > You should compare $s$ with the scale factor employed after tuning (i.e. the maximum learning rate when a schedule is used), which is usually smaller than 0.1 as you observed. One might hope that mechanic would also learn to produce a schedule as well, but in practice including the schedule seems better (and our theorem only shows that it can find the single best scale for any given schedule, not that it will find an entire schedule).

---

### Official Review · Reviewer_mCwi · 2023-07-09

**Soundness:** 3 good
**Presentation:** 3 good
**Contribution:** 3 good
**Rating:** 6
**Confidence:** 4

**Summary:**

This paper proposes a parameter-free technique to tune the learning-rate scale factor automatically, which can be applied to any given base optimization algorithm to match its performance given carefully tuned hyper parameters. This approach is mainly empirical in nature, though grounded in reduction from recent advancements in theoretical understanding of online convex optimization. The implementation is also "scale-free", i.e. invariant to scaling the stochastic gradients $g_t$ by a constant. The authors provide experimental evidence to support their claims, including both training from scratch as well as transfer learning, and across domains and model architectures.

**Strengths:**

With the rapid rise in both demand and costs to train large models, the compounding effects of hyper-parameter tuning transitioned from being a technical inconvenience to a true bottleneck. While there have been some advancements in parameter-free optimization, any practical contribution to this field is highly valuable. A general-use, easy to implement extension to existing optimizers that act as a force multiplier and improves upon their results falls comfortably into this category.
In addition to providing some theoretical justification to their method, the authors also provide a detailed, easy-to-follow background to the field and its main results thus far. The paper also include some interesting intuitions by the authors, instead of post-hoc explanations of empirical findings.
The experiments performed by the authors include various domains and model architecture, and notably include both pre-training from scratch as well as fine-tuning which are known to behave differently.

**Weaknesses:**

As stated by the authors themselves: their "primary contribution is not new theoretical development, but instead the translation between theory and practice". Since this paper is focused mainly on empirical evidence rather than new theoretical results, the burden of proof grows larger. While the authors do perform various experiments, there is key information that is missing to fully convince in the efficacy of the approach.
1. The experimental setting is lacking some details. Namely, how were the hyper-parameters for the base algorithm chosen? Following existing procedures while altering some hyper-parameters (e.g. dropping the NSP loss) may affect the results. Additionally, the given hyper-parameters are not guaranteed to be optimal in the first place. To make the claim that MECHANIC helps in achieving competitive results to carefully tuned optimizers, the base optimizers should indeed be carefully tuned.
2. Moreover, statistical significance of the results is not assured, as no measure for variance in the results is given. Very often, the advantage of the MECHANIC based training is small enough to potentially be attributed to the inherent noise in the evaluations.
3. It is not clear that all models converged in the presented results as there are missing details. For example, Table 1 provides fine-tuning results of BERT like models on GLUE, which underperform the results reported by Devlin et al. 2018, and Liu et al. 2019. This may be due to the difference in pre-training corpus, or due to undercutting of the models. In the latter case, additional training may have changed the results.
4. There are missing experiments to show how well does MECHANIC operate when the hyper-parameters of the base algorithm are not well tuned. If it fails in such cases, then one still has to tune the parameters of the base algorithm, and gain nothing by using this approach.
5. there are missing ablation tests - what is the effect of $\lambda$ and the effect of $\beta$? Also, what is the effect of learning rate schedulers on the performance of MECHANIC?
6. While the paper focuses on empirical findings, the authors suggest the method should be robust, but with no guarantees. For example, line 194-196 the authors claim that weight decay should not affect the operation of MECHANIC, but provide no proof (lie 195-196) and no empirical evidence.
7. While the method is presented as parameter-free, there is evidence that tuning its hyper-parameters may indeed be required. For example, lines 175-176 discuss the importance of setting $\beta$. While the authors try to obviate the need to tune it, the choice of defaults may not generalize well to other settings. Also, lines 239-240 mention that tuning $\lambda$ may be beneficial.


Some additional comments regarding the presentation of the paper:
* Algorithm 2 is referred to quite a lot and is being analyzed in the paper. As such, I think it will be beneficial for the reader if it is included in the body of the paper.
* In line 49, you state the "Empirical studies often take the route of "hyper-gradient descent". However, this approach is rarely used in practice, and even in the work you cite, most empirical approaches do not follow it, nor compare to it.
* The "scale" buy which the paper measure gains is not consistent. When MECHANIC is outperformed by base algorithms it is said to be "quite competitive", while when it has the advantage by smaller absolute gains, it is said to outperform. For example, see lines 247-248 which discuss the results of Figure 2.


There are a few typos and styling issues that should be fixed prior to final submission:
1. line 38, there is a missing "to" between "allowed" and "make"
2. line 93, the words "let and" were swapped
3. line 113, **missing key information** - the sentence of how you set the scaling factor is missing "and so by setting <MISSING> To show..."
4. In Table 2, there are at least 5 cases where the base algorithm achieved **identical** results to those of the MECHANIC tuning, while the table gives only the results of the MECHANIC tuning in boldface which is confusing and misleading.
5. line 218, "works" $\rightarrow$ "work"
6. line 219, "not that" is swapped
7. The bibliography style is inconsistent in its presentation. For example, [7,11,14,33] are all cited from CoLT, each with different styling. [20,21,36] are both from NeurIPS but with very different styles as well. Also 35 cites from arXiv differently from e.g. 38.
8. All references in the paper are not "clickable". Namely, figures, table and citation references are not clickable in the PDF

**Questions:**

1. How were the hyper-parameters of the base algorithms set? How are the results change if the hyper-parameter of the base algorithm are under-optimal? Was there a thorough tuning phase to verify the chosen hyper-parameters (including the total number of update steps) of the baseline algorithm were indeed correctly tuned?
2. What is the standard deviation in the given results?
3. What is the effect of $\lambda$ and the effect of $\beta$ ? Also, what is the effect of learning rate schedulers on the performance of MECHANIC?

**Limitations:**

The authors have addressed the limitation of their work adequately. Namely, the authors mention the lack of theoretical guarantees or intuition into some of the results, as well as potential future work. There is no ethical considerations relevant in this case.

---

> ### Author Rebuttal · Authors · 2023-08-10
>
> Thank you very much for your detailed comments! Your feedback will be very helpful to us in revising the paper. In the below we provide some more detail that we hope will address your main concerns.
>
> **Main concern about missing empirical information**
>
> We provide a lot more detail in the appendix section B. However, to specifically answer your questions:
> ```Namely, how were the hyper-parameters for the base algorithm chosen?``. We have actually tried to be faithful to the original baseline whenever possible. For instance, for our image classification experiments, we even use the exact code open-sourced by the original authors for both pre-training and fine tuning, with original hyperparameters (including the tuned learning rate) to limit any confounding factors. There is exactly one place where we deviate from the baseline, which is the removal of NSP loss. We did that only to simplify our experiments since it did not affect the performance of the baseline at all.
>
> ```To make the claim that MECHANIC .. the base optimizers should indeed be carefully tuned.```: You are absolutely right. Since learning rate is the most significant hyperparameter for this study, for all baselines we either grid sweeped over a reasonably large range learning rates (e.g. BERT) or (to limit costs) used a well tuned learning rate from a paper on the code open-sourced by the authors of the baseline (e.g. ViT pre-training) Exact values can be found in sections B.1 and B.2. We also note that as a sanity check, we report competitive results on the *baselines* when compared with what was reported by the original authors.
>
> ```Moreover, statistical significance of the results is not assured```: For results on small dataset like GLUE, you are right, it is hard to attribute the gains directly to Mechanic since there can be a lot of variance. But this is true even for the baselines. For that reason, similar to what was done in the original baseline papers, for all finetune runs, we report an average of 3 runs to counteract the variance. Finally, our aim is not to claim that with Mechanic one can get better performance than the baseline with well tuned learning rates, we merely hope to illustrate that it can help eschew the need for learning rate tuning for similar performance.
>
> ```It is not clear that all models converged```: We are a little confused by this concern. We did look again at the original BERT results and compared our ```baseline``` results with what they report. We would like to note that contrary to the concern, for some GLUE datasets such as QNLI, SST-2 ad QQP, with BERT-L, we in fact report better or almost the same results than what was reported with BERT on dev set (from Table 5 in Liu et al). Let us know if we misunderstood your concern.
>
> ``` There are missing experiments ..``` This is an interesting question and your point is well taken! We presented the case in which the other hyperparameters of the base algorithm (such as the schedule) are well-tuned. That said, we have observed that Mechanic continues to improve upon even poorly tuned baselines although we did not perform enough experiments on this case for the original submission as we felt that the well-tuned case was the one of most practical significance. Our hope is that Mechanic is indeed robust against other changes in hyperparameters, but there are still cases where Mechanic is currently less effective, such as when using dropout or normalized gradient updates like LARS and LAMB. These are certainly limitations of Mechanic and we will make sure to document them as such in our limitations section.
>
> ```\lambda, \beta and lr schedules```:  [Also elaborated in response to reviewer 2EmS] you are absolutely right that ablating hyperparameters may result in useful insights, however in this case, beta (with the addition scheme), s_init and \lambda (if not too big) are robust enough not to produce meaningful difference in performance. Regardless we are happy to add the plots if you think they help!
>
> Inspired by your suggestion, one interesting ablation study might be to change num_betas (*n* in Alg 1), following table shows  *n* vs Acc of ViT-B/16 on JFT-300M:
>
> ```
> | n   | 2    | 4    | 6    | 8    |
> | Acc | 48.9 | 49.5 | 49.9 | 49.6 |
> ```
> Interestingly,  training runs with *n* < 6 perform worse and also shows instabilities, however, increasing *n* beyond 6 leads to slightly poor performance.
>
> **Tuning other hparams**
>
> ```If it fails in such cases, then one still has to tune the parameters of the base algorithm…```: We focused here on tuning only the scale factor for the updates, NOT in particular regularization constants or the schedule or any other aspect of optimization. Tuning this scale is typically one of the most important hparams, so removing the need to tune it is already a significant improvement that will certainly save computational resources. Note that without mechanic you still need to tune those parameters anyway. That said, we certainly acknowledge that our method does not completely obviate the need for ALL tuning.
>
> ```While the method is presented as parameter-free``` Please note that for our experiments we used the SAME values of beta/lambda across various models and datasets, providing evidence that in future tasks no alteration/tuning of mechanic is necessary. In the case of beta, the use of multiple beta values at once does actually come from some theoretical motivation for why this technique obviates the need to tune beta. For lambda we have less understanding and simply propose it as an empirical improvement. In practice it seems the default value is good, but certainly you could choose to tune it if you desire.
>
> The sentence on line 113 should have ended after the parentheses, so the words “and so by setting” should have been deleted. The setting for $\mathring{s}$ in this particular example is provided later in the paragraph. See also our response to reviewer 7tau for a proposed alternative phrasing of this paragraph.

---

> > ### Comment · Reviewer_mCwi · 2023-08-16
> > **Thank you for your clarifications. Some follow-up questions.**
> >
> > Thank you for your clarifications.
> >
> > Regarding the BERT/RoBERTa results: The results for BERT provided in Table 1 of Devlin et al. 2018 and Table 5 in Liu et al. 2019 are on the test set, and thus are expected to be lower to begin with. Furthermore, since the primary difference between RoBERTa and BERT is the addition of more training data (book corpus), the removal of the NSP loss (which was shown to improve results) and most importantly longer training, MECHANIC results should be best compared to RoBERTa’s validation results as presented in Table 8 of Liu et al. In every instance, RoBERTa outperforms your baseline, leading to my concern that the models may not have fully converged.
> >
> > Regarding the missing experiments, you mentioned that for the baselines, you used well-tuned optimizers (either from the original authors or by performing a grid search). My concern here is the hyperparameters used for the model trained with MECHANIC. If they aren't calibrated correctly from the outset and the result is subpar, then one would need to perform hyperparameter tuning for the base optimizer in any case. Given that many of the experimental results don't show performance enhancement but rather produce results on par with well-tuned models, I wonder about the benefits of using MECHANIC if one still needs to perform this tuning.

---

> > > ### Author Response · Authors · 2023-08-16
> > > **answer to follow-up**
> > >
> > > Thank you for taking a look at our response promptly! We would be more than happy to provide additional clarification.
> > >
> > > ```Regarding the BERT/RoBERTa results```: We are again confused. We are looking at the first row of Table 5 from Liu et al. They mention that those results are **Single-task single models on dev** and not the test set. This is the exact setting that applies to our results too. We are comparing our results with **BERT-Large** (first row) from that table.
> > >
> > > ```Regarding why Roberta is not our baseline```: Note that we removed the NSP loss but the similarity of our setting to Roberta ends there. Most notably, we do **not** use 1) dynamic masking 2) Full doc sentences (bert uses short sentence pairs and we do too) 3) training for longer (like you mentioned)  and much larger batch size (with differences in adam optimizer). These changes may explain the difference in performance. You are right that comparing to Roberta’s setting may also be a fruitful evaluation of Mechanic but we hope that this does not discredit our results on vanilla BERT (which is also well understood and a lot of optimization papers use as a baseline e.g. LAMB [1])
> > >
> > > ```Regarding model not being fully converged```: If your concern is whether we have linear decayed learning rate for the baseline to convergence, then yes we did! If you are asking why we didn’t train for as long as Roberta (or even longer), we would like to emphasize that pre-training runs are **very** expensive and we do train for a fair comparison with a well-known baseline of vanilla BERT for meaningful results. Additionally, if you are interested, we compare Mechanic on variety of other benchmarks too, most notably in figure 2, we compare on IWSLT14 (LSTM) and BookWiki (GPT Transformer) in the language domain.
> > >
> > > ```Regarding calibration of mechanic hyperparameters```: We don’t follow the concern here. Mechanic is designed to remove the need to tune the learning rate scale factor and ONLY that need. Of course other hyperparameters (such as a schedule or batch size) might still need to be tuned when using mechanic. However, even just removing the need to tune the learning rate scale factor results in a significant saving as this is usually one of the most critical hyperparameters to get right. In fact, since our experiments used only the tuning from the BASE algorithm for these other hyperparameters, the performance of mechanic would only increase if we were to actually tune for mechanic. We mostly opted not to do this so as to illustrate that mechanic’s good performance is due to the automatic scale factor tuning and not some interaction with other hyperparameters.
> > >
> > >
> > > [1]: https://arxiv.org/abs/1904.00962

---

### Official Review · Reviewer_7tau · 2023-07-15

**Soundness:** 3 good
**Presentation:** 1 poor
**Contribution:** 4 excellent
**Rating:** 7
**Confidence:** 3

**Summary:**

The paper proposes a scheme to automatically tune the learning rate scale factor for any gradient-based optimization algorithm.  The method can be viewed as a practical realization of recent theorertical results in online convex optimization (OCO), which reduces the problem to minimizing the regret of a one-dimensional OCO algorithm. The authors test their proposal on a range of large-scale deep learning benchmarks, showing  competitive performance compared to strong manually tuned baselines. It is also shown that the method can outperform a recently proposed learning rate tuner, D-adaptation, without requiring the modifications accross base optimizers that D-adaptation does.

**Strengths:**

* Strong empirical evaluation, which shows solid results against strong baselines in large-scale contexts where learning rate tuning is known to play a criticial role.

* Great potential  for practicaly applicability, as it provides an off-the-shelf  wrapper that can be used on top of any gradient-based optimization algorithm.

* (At least part of) the approach is theoretically motivated -- and subject to theoretical analysis.

Given the above, I believe the project can be turned into a very strong submission -- provided the weaknesses below are addressed.

**Weaknesses:**

Despite the paper's great potential, a major reservation I have, which justifies my rating, is about the clarity of the presentation -- at least for those without strong expertise of the relevant literature.

* The background section, which motivates the approach, is based  on prior work -- specifically, recent progress in parameter-free online convex optimization (such as ref [28]). I think the authors can do a much better job at reproducing and explaining these known results (the fact that presentation is plagued with misprints does not help).

For example, while Theorem 1, which bounds the regret of Mechanic in terms of the regrets of Base and Tuner, is relatively clear, the exposition of how it is exploited (end of Section 2.1 and Section 2.2 ) is very unclear to me. To be specific, the right-hand-side in the last equality on line 97, s dot shows  up in both regrets, which suggests some sort of tradeoff between both regrets --  I do not follow the reasoning  that the problem is completely relegated to Tuner (for example, the optimal s dot  on line 101 from the BASE regret term looks distinct from the optimal s dot on line 119 from the tuner regret).  Furthermore,  the paragraph between l112-l122 completely lost me.

 I think this would be highly beneficial to rewamp entirely this exposition  for the sake of readability,  especially for non OCO experts like myself.(I formulate more specific comments / questions in Questions section.

* In Section 3 the specific form of Tuner (even  in its simplified form after line 148) seems to come out of the blue.
I am not sure whether or not it was supposed to follow from the background analysis; if it does, I think being  explicit  about  the transition would be helpful (of course the empiricial justifications of some specific ingredients, as in 3.2 for weight decay, is completely fine).

**Questions:**

I am confused by the notation (line 41 or line 95) suggesting that the MECHANIC iterates are mere interpolations of the initial point and the BASE iterates.  It makes it look like one needs to run the base algorithm  once completely to be able to define the MECHANIC iterates, which is clearly not the case from Algorithm 1. I see how  MECHANIC depends on the BASE *updates* -- but these are built from  gradients that depend on previous MECHANIC iterates, not on the BASE iterates.  (The authors seem to acknlowedge this abuse of notation in Footnote 1, but I cannot see the mathematical justification that the footnote is referring to).


I am not sure about the editorial choice that  consists of motivating the algorithm with the OCO analysis.
I feel a better choice could be to first introduce the wrapper as an additional  optimization step over the scaling factor $s$ with loss $\ell_t(x_1 - s \sum_t \mbox{base updates}_t)$, which I find intuitive and elegant in and of itself  -- as it optimizes in the direction of what would have been a good scale in hindsight  ; then to introduce OCO subsequently for the sake of the analysis.

Miscalleneous:

* l38 allowed make <-- allowed to make

* Footnote 1: presently <-- below or shortly

* Inconsistent index for s_t between l41 and l42.

* From l45, I suggest to just create a new "related work" section. An explicit list of contributions in the introduction would also be helpful.

* l93 let and <-- and let

* l103 approximates of the gradient <-- approximates the gradient

* l110: isn't an example of such an advanced algorithm SGD initialized a s=0? (from l83).

* l110: A TUNER subscript is missing in the regret formula.


* l113 missing words: "so by setting To show" ...


* On Algorithm 1 : l6 gk <-- gt. I suggest adding What is zt? x_ref not updated?

* Table 1: it would be nice to see standard deviations reported as well

* The last mentioned limitation in section A looks like an advantage to me (no validation set required).

**Limitations:**

Limitations adequately acknowledged in the appendix.

---

> ### Author Rebuttal · Authors · 2023-08-10
>
> Thanks very much for your detailed review and feedback on the presentation - making sure everyone can follow the paper is a priority for us. Unfortunately, we cannot provide an updated revision, so below we’ve copied in some changes that will ameliorate the issue. We would be happy to hear your feedback, and we intend to make a careful edit of the paper with an eye to accessibility.
>
> ```For the paragraph starting at line 99, we propose to add detail as follows:```
>
> “
> With this result, finding the optimal $s$ can usually be completely relegated to $\texttt{tuner}$. Although the value $\mathring{s}$ appears in both terms of the sum $\text{Regret}\_{\text{linear}}^{\texttt{tuner}}(\mathring{s}) + \mathring{s}\text{Regret}\_{\text{linear}}^{\texttt{base}}(x^{\texttt{base}}\_1+(\mathring{x}-x^{\texttt{base}}\_1)/\mathring{s})$, it turns out that for essentially all known base algorithms, there is a particular value $\mathring{s}$ that causes $\mathring{s}\text{Regret}\_{\text{linear}}^{\texttt{base}}(x^{\texttt{base}}\_1+(\mathring{x}-x^{\texttt{base}}\_1)/\mathring{s})$ to obtain the optimal regret bound $\|\mathring{x}\|\sqrt{\sum_{t=1}^T \|g_t\|^2}$. This value for $\mathring{s}$ is unknown a priori, and depends on the data and the base algorithm. However, if $\text{Regret}\_{\text{linear}}^{\texttt{tuner}}(\mathring{s})$ is sufficiently small for this unknown value $\mathring{s}$, then overall we achieve the optimal regret bound without having to know $\mathring{s}$ ahead of time. Note that setting $\mathring{s}$ in this way can be done entirely in the analysis without modifying the algorithms, as justified by the infimum in the Theorem.
> “
>
> ```For the paragraph at line 112:```
>
> “
> In the a theoretical development of this technique, it is necessary to prevent the terms $\langle g_t, x_t^{\texttt{base}}  - x_t^{\texttt{base}}\rangle^2$ from becoming too large (as otherwise $\text{Regret}^{\texttt{tuner}}$ is too large). Typically, this is accomplished by constraining the base algorithm to satisfy $\|x_t^{\texttt{base}}  - x_t^{\texttt{base}}\|\le \rho$ for some user-specified arbitrary $\rho$. Enforcing such a constraint means that the regret bound (2) would only apply to $\|\mathring{x}\|\le \rho$, but ensures that $\langle g_t, x_t^{\texttt{base}}  - x_t^{\texttt{base}}\rangle^2\le \rho^2 \|g_t\|^2$. Thus, by setting $\mathring{s} = \|\mathring{x}−x_1^{\texttt{base}}\|/\rho$, the combined algorithm obtains the optimal regret bound of $O(\|\mathring{x}−x_1^{\texttt{base}}\|\sqrt{\sum_{t=1}^T \|g_t\|^2})$ (amazingly, the value of $\rho$ is irrelevant!). In practice however, we do not attempt to explicitly enforce any such constraints and simply rely on the intuition that any non-diverging algorithm is unlikely to produce excessively large iterates.
> “
>
> ```Regarding motivating the Tuner algorithm: We will first motivate a further simplified update with the following text:```
>
>
> “
> While the specific form of our tuner update is based upon more involved analysis, we can capture some intuition by appealing to the familiar SGD algorithm. First, notice that the “gradient” sent to Tuner is $h_t$. Thus, the SGD update would be $s_{t+1} = s_t - \eta h_t$ for some learning rate $\eta$. It turns out that the analytically optimal value for $\eta$ is $\frac{\mathring{s}}{\sqrt{\sum_{i=1}^T h_i^2}}$. Unfortunately, we do not know this value at first. Instead, at time $t$ we estimate the denominator with the running sum $\sqrt{\sum_{i=1}^t h_i^2}$ and the numerator with the “optimistic” approximation $\mathring{s}\approx s_t$, so that the update becomes $s_{t+1} = s_t - \frac{s_t h_t}{\sqrt{\sum_{i=1}^t h_i^2}}$. Unfortunately, this update is unstable and can result in exponential blowup in the $s_t$ values as bigger $s_t$ results in bigger learning rates. Much of the technical detail in our update is designed to deal with this instability, essentially by introducing a carefully designed decay into the $s_t$ update.
> “
>
>
> ```Regarding the intuition of taking the “gradient with respect to s” of $\ell(x^{\texttt{base}}_1 + s\cdot(x^{\texttt{base}}_T -x^{\texttt{base}}_1)$```: We agree this makes sense a high level, but in fact the analysis is totally different and we could not see any way to formally justify the algorithm with this intuition. We were a bit wary of emphasizing it too much since we cannot be confident that variations based solely on this intuition will behave well. Nevertheless, we will flesh out the idea a bit more in the discussion after Theorem 1. This is actually related to your question about footnote 1 - our analysis is essentially a mathematical “trick” that enables us to sidestep issues that arise with more intuitive arguments.
>
> ```Regarding footnote 1, as well as the concern about needing to first do a run of the base algorithm```: The justification is Theorem 1, and you are correct that we do NOT need to run the base algorithm first.
> Our analysis is based on regret and actually does not stipulate any particular meaning for the vectors $g_t$. The base algorithm is viewed as a black box taking as input any sequence of vectors $g_1,\dots,g_t$ and yielding some $x_{t+1}^{\texttt{base}}$. Although it is intuitive to set $g_t=\nabla f(x_t^{\texttt{base}},z_t)$, this is not assumed in our analysis. Nevertheless, all standard algorithms like SGD, AdaGrad etc ensure bounded regret for arbitrary sequences $g_t$.
>
>
> ```Other questions:```
> Line 110: The “advanced” part of these algorithms is that they obtain the bound $\mathring{s}$ automatically, while SGD would require tuning the learning rate depending on $\mathring{s}$.
>
> Line 113: The partial phrase “so by setting” should be deleted: the sentence ends after the parentheses (the value for $\mathring{s}$ is provided later in the paragraph).
>
> Algorithm 1: $z_t$ represents the $t^{th}$ minibatch (see line 11). $x_{\text{ref}}$ should be $x_1^{\texttt{base}}$ (and so does not need updating).

---

> > ### Comment · Reviewer_7tau · 2023-08-21
> >
> > I thank the authors for their rebuttal and theclarifications, which will indeed be very welcome in the revision. Since clarity and presentation was my only major concern, for what I think is otherwise a strong submission, I've raised my score.

---

### Official Review · Reviewer_2EmS · 2023-07-22

**Soundness:** 2 fair
**Presentation:** 3 good
**Contribution:** 3 good
**Rating:** 6
**Confidence:** 3

**Summary:**

This paper develops a method to automatically determine the learning rate of an optimization algorithm. Their approach, referred to as Mechanic and motivated by theoretical advances in online convex optimization, determines a new learning rate for iteration t using an update that resembles an adagrad update plus an additional decay factor. Their method is tested in deep learning settings ranging from language to large scale vision, where it approaches and sometimes outperforms a "tuned baseline".

**Strengths:**

This paper has many strengths:
- The problem that the authors explore is very important to the community. Even if their method is not a universal solution, any sound progress towards this target is potentially valuable. If the community succeeds in removing the need for tuning learning rates, significant resources use will be eliminated.
- Mechanic can be applied to any base algorithm, not only in theory but also in practice: the authors show results for Adam, Lion, and SGD.
- Mechanic is tested in various different domains from masked language modeling to large scale JFT pre-training.
- Mechanic has only two tunable hyper-paramters, s_init and the vector of beta.

**Weaknesses:**

- The central weakness in this paper is the lack of clarity surrounding the "tuned baseline". How was that baseline tuned? What scheduler is used? Is it the best learning rate and if so out of how many searched? It would be great to see a line plot of LR vs. perf for the baseline and then a horizontal line for mechanic. This weakness is why I've listed the soundness as only fair.
- There are no ablation studies for the relevant hyperparameters s_init and the betas.
- Minor: There are some claims in the paper which I do not believe to be true. For instance, in the conclusion the authors claim that it is infeasible to manually tune a scale factor for each layer. However, there is work, e.g.,  mu-transfer which aims to specify a "corrected" learning rate for each layer.

**Questions:**

- Different base algorithms and problem settings result in different performance for Mechanic vs. the baseline. Do the authors have any ideas why this may be?
- How is performance affected by s_init and the betas.
- I was slightly lost in the explanation for how multiple betas were used in the section where the adagrad like update transforms into more of an adam like update. Could the authors extend this trick so that multiple betas could be used simultaneously in other contexts, e.g., for the adam estimators for the params?
- Can you explain the tuning that went into the baseline. Ideally a plot of LR vs. perf for the baseline could be contrasted with mechanic.
- Why do the authors compare only to D-adapt and not, e.g., DoG https://arxiv.org/abs/2302.12022.

**Limitations:**

Yes, the limitations are discussed.

---

> ### Author Rebuttal · Authors · 2023-08-10
>
> **Main concern about baseline hyperparameter tuning**
>
> We provide a lot more detail into the appendix section B. However, to specifically answer your questions:
>
> ```How baseline was tuned and Is it the best learning rate and if so out of how many searched```: For all baselines we either grid sweeped over a reasonably large range learning rates (e.g. BERT) or (to limit costs) used a well tuned learning rate from a paper on the code open-sourced by the authors of the baseline (e.g. ViT pre-training) Exact values can be found in sections B.1 and B.2
>
> ```What scheduler is used```: Assuming you mean LR scheduler, we used the schedules mentioned in the original baselines for both with and without Mechanic. Specifically, linear schedule for BERT and cosine schedule for ViT. Note that Mechanic does not eschew a need for a schedule (at least not yet), it simply learns the right scale factor to be used for a predefined schedule.
>
> ```It would be great to see a line plot of LR vs. perf for the baseline and then a horizontal line for mechanic.```: This is a great suggestion! We will post data for this during the discussion period and even add plots in the final version of the paper.
>
> **Regarding betas and s_init**
>
> You are absolutely right that ablating hyperparameters may result in useful insights, however in this case, both beta (with the addition scheme) and s_init are robust enough not to produce meaningful difference in performance.
>
> We have found that using a single value for beta required tuning of the beta value for each model/dataset separately. However, the “addition scheme” we used does not. This was motivated by theory (via the “combining regret bounds via addition” technique introduced by [33]). We found that as long as a reasonable beta value for the problem is present in the betas sequence (default), performance doesn’t change.
>
> Regarding tuning of s_init, we intended the scheme to work with “obviously too small” values like 1e-8 (a property that is supported by the logarithmic dependence on 1/s_init in theory) but in practice changing s_init to even a few orders of magnitude does not affect performance much.
>
> Let us know if you still want us to add those results in the paper, we would be more than happy to do so!
>
> **Regarding the multiple beta trick**
> The trick we used to combine multiple beta values in the tuner algorithm relies on a theorem relating to parameter-free online algorithms that states a “meta algorithm” created by adding the outputs of many sub-algorithms will have performance no worse than the best of the sub-algorithms (even if we do not know which one is best at first). Thus, since we did not know which beta value would be best at first, we simply added the outputs of many instances of Tuner with different beta values. In principle, one might be able to apply a similar trick with the beta values in a high dimensional algorithm like Adam, but this would require blowing up the memory and runtime of Adam by a factor of [number of beta values]. Since Tuner requires only O(1) memory and time, this blowup is irrelevant compared to the time needed to simply compute a gradient, but Adam requires O(d) time/memory where d=[number of parameters in the model], so this may incur more significant performance issues.

---

> > ### Comment · Reviewer_2EmS · 2023-08-12
> > **Thanks for your response.**
> >
> > I look forward to seeing the data for the plot that you mention, and think it will be a good contribution to the paper.

---

> > > ### Author Response · Authors · 2023-08-15
> > > **Thank you and rebuttal round #2**
> > >
> > > Absolutely! Here are some additional data points that you requested. We hope that this resolves your concerns. Please let us know if you need any other information.
> > >
> > > **Plots of learning rate vs performance of BERT models**
> > >
> > > ```
> > > Adam Optimizer
> > > | LR  |        | 5e-4 | 1e-3 | 2e-3 | 5e-3 | 1e-2 | Mechanic
> > > ---------------------------------------------------------------
> > > | Acc | BERT-B | 71.1 | 71.5 | 71.5 | 71.5 | 71.3 | 71.7
> > > | Acc | BERT-L | 75.0 | 75.4 | 75.4 | 74.6 | 74.4 | 75.3
> > > ```
> > >
> > > ```
> > > Lion Optimizer
> > > | LR  |        | 5e-5 | 1e-4 | 2e-4 | 5e-4 | 1e-3     | Mechanic
> > > ---------------------------------------------------------------
> > > | Acc | BERT-B | 70.8 | 70.8 | 71.1 | 71.8 | 71.4     | 72.0
> > > | Acc | BERT-L | 75.1 | 75.6 | 75.7 | 74.7 | Diverged | 75.5
> > > ```
> > >
> > > **Ablations of s_init, \lambda and num_betas**
> > >
> > > Since you were interested, we also re-ran the ablations of s_init, \lambda and num_betas using ViT-B/16 on JFT-300M with *M*-Adam . Definitely let us know if you require any other information.
> > >
> > >  ```
> > > | s_init | 1e-8 | 1e-7 | 1e-6 | 1e-5 | 1e-4
> > > ---------------------------------------------------------------
> > > | Acc    | 49.8 | 49.8 | 49.9 | 49.7 | 49.6
> > > ```
> > >
> > > As we expected from theory, changing s_init by even orders of magnitude does not result in much difference in performance.
> > >
> > >  ```
> > > | wd (or \lambda) |  0   | 1e-3 | 1e-2 | 1e-1 | 1e0
> > > ---------------------------------------------------------------
> > > | Acc             | 49.7 | 49.8 | 49.9 | 49.7 | Diverged
> > > ```
> > >
> > > We have observed that while \lambda is helpful in stabilizing Mechanic on some problems, as long as it is set to a reasonable value it does not affect performance by a lot.
> > >
> > > ```
> > > | num_betas |  2   |  4   |  6   |  8
> > > -----------------------------------------
> > > |    Acc    | 48.9 | 49.5 | 49.9 | 49.6
> > > ```
> > >
> > > Interestingly, we find that having training runs with num_betas < 6 not only perform worse but also leads to instabilities throughout the training run (we will include plots in the final paper). However, increasing num_betas beyond 6 does leads to slightly worse performance.

---

> > > > ### Comment · Reviewer_2EmS · 2023-08-16
> > > >
> > > > Thanks for the results. I've raised my score. I am wondering, how do you know it's not a matter of the number of betas but rather what the betas are?

---

> > > > > ### Author Response · Authors · 2023-08-17
> > > > > **Number of betas vs betas**
> > > > >
> > > > > Thank you so much for increasing your score and supporting our work!
> > > > >
> > > > > ```Are you asking about num_betas ablation results?```
> > > > >
> > > > > You are right that there may be an optimal single beta value which can recover the best results when using multiple betas but we've found that single optimal beta is problem dependent and needs to be tuned. Thus the main advantage is actually just the fact that we don't have to tune it. Since there is not much cost to adding more beta values we specify the end of the range and don't tune the start of the range. For instance, num_betas = 2 means we use betas = [0.9, 0.99] or num_betas = 4 means betas=[0.9, 0.99, 0.999, 0.9999].  This allows mechanic to do as well as any beta in the list. Of course, the “truly optimal” value may be something more like 0.65 and so not be in the list, but we expect choosing the correct order of magnitude to be reasonably good, similar to how a grid search over learning rates will miss the exact optimal rate but still perform well.
> > > > >
> > > > > Please let us know if you had more questions and we hope that you would consider strongly supporting our work and increase the score to "Accept".

---

### Decision · Program_Chairs · 2023-09-21

**Decision:**

Accept (poster)

**Comment:**

The reviewers have reached a consensus in favor of accepting this paper, appreciating the way it bridges theory and practice and provides promising empirical results. However, the reviewers have expressed important concerns about the clarity of the experiments and missing ablations. I fully agree with the reviewers on both counts, recommending acceptance while **requiring** the authors to revise the paper by adding all the explanations and additional experiments discussed with the reviewers. Crucial details such as the methodology for hyperparameter selection must appear in the main part of the paper rather than the appendix.

In addition to all the important points raised by the reviews (which the authors should carefully peruse when making the revision, as some comments remained unaddressed), I have a few comments of my own:

1) The submitted paper and supplement appear to be missing the following crucial detail: **what LR is used in the BASE optimizers**? This matters, since TUNER is not invariant to the BASE LR (as evidenced by the fact that LR schedules matter). The authors’ reply to reviewer DR7W suggests that the LR for all BASE algorithms is 1.0. While this seems like a fair setting, if taking BASE’s LR to be 1.0 is essential it would contradict the paper's claim that BASE could be any algorithm. The authors must discuss the choice of BASE’s LR in the main part of the paper. In addition, they should add an experiment testing whether the LR parameter in the BASE algorithm truly does not matter as claimed.

2) As reviewer 7tau briefly notes, deviation metrics are missing throughout the paper. At least for the fine-tuning experiments, the authors must repeat them with multiple seeds and report, e.g., standard deviation along with the accuracy.

3) The authors call a JFT-300M a “popular” dataset, which I find highly objectionable considering it is available only to Google employees, while the rest of the community does not even have a proper description of the content of the dataset. Remove this misplaced adjective.